# FOSSIL: Regret-Minimizing Weighting for Robust Learning under Imbalance and Small Data

[anonymized]

## Abstract

Imbalanced and small-data regimes are pervasive in domains such as rare disease imaging, genomics, and disaster response, where labeled samples are scarce and naive augmentation often introduces artifacts. Existing solutions—such as oversampling, focal loss, or meta-weighting—address isolated aspects of this challenge but remain fragile or complex. We introduce FOSSIL (Flexible Optimization via Sample-Sensitive Importance Learning), a unified weighting framework that seamlessly integrates class imbalance correction, difficulty-aware curricula, augmentation penalties, and warmup dynamics into a single interpretable formula. Unlike prior heuristics, the proposed framework provides regret-based theoretical guarantees and achieves consistent empirical gains over ERM, curriculum, and meta-weighting baselines on synthetic and real-world datasets, while requiring no architectural changes.

## 1 Introduction

Modern machine learning systems are increasingly deployed in high-stakes domains such as healthcare, finance, and safety monitoring, where decision-making must remain reliable under *severe class imbalance*, *limited data*, and *noisy augmentation* (He & Garcia, 2009; Johnson & Khoshgoftaar, 2019; Krawczyk, 2024; Yang & Chen, 2023; Wu & Liu, 2023). In such settings, a single misclassification—e.g., missing a rare disease or overlooking a fraudulent transaction—can lead to catastrophic outcomes. Yet, despite progress in optimization and representation learning, current training pipelines remain vulnerable to *augmentation dominance* and fail to *adaptively adjust sample importance* throughout training (Shorten & Khoshgoftaar, 2019; Zhang et al., 2022).

Curriculum learning (Elman, 1993; Bengio et al., 2009; Guo et al., 2018; Wang et al., 2021) offers a principled mechanism to organize training by difficulty, leading to faster convergence and better generalization. However, existing curricula typically ignore two critical aspects: (1) imbalanced distributions, where minority classes must be up-weighted, and (2) augmentation bias, where synthetic samples may overwhelm real data and cause overfitting (Chen & Zhang, 2022; Wu & Liu, 2023). Recent reweighting approaches (Ren et al., 2018; Shu et al., 2019; Cui et al., 2019; Yang & Chen, 2023) partially address imbalance, but lack a unified framework that integrates class rarity, sample difficulty, augmentation awareness, and training dynamics. We introduce *FOSSIL* (**F**lexible **O**ptimization via **S**ample-**S**ensitive **I**mportance **L**earning), a simple yet powerful weighting strategy for robust learning under imbalance and augmentation. The core idea is captured by a single formula:

$$w_i(t) \;=\; \underbrace{\frac{1}{K\,p(y_i)}}_{\text{class term}} \cdot \underbrace{\exp\!\left(-\frac{d_i}{T_t}\right)}_{\text{difficulty term}} \cdot \underbrace{\left(1 - \gamma_t\,\mathbf{1}\{i \in \mathcal{A}\}\right)}_{\text{augmentation penalty}} \cdot \underbrace{\min\!\left(1, \frac{t}{t_{\text{warm}}}\right)}_{\text{warmup term}}, \tag{1}$$

which dynamically balances minority-class weighting, curriculum progression, and augmentation penalties while stabilizing early training via warmup.

The framework is theoretically grounded: we establish regret guarantees and show it subsumes curriculum learning, focal loss, and class-balanced weighting as special cases. Our analysis draws on

online convex optimization and regret minimization (Shalev-Shwartz, 2012; Hazan, 2016b), connecting FOSSIL to a broader theoretical foundation. Empirically, we show that our method consistently outperforms ERM, curriculum, and meta-weighting baselines on both synthetic imbalance tests and real-world medical imaging datasets. This paper makes three main contributions. First, we introduce a unified importance-learning framework integrating class imbalance handling, difficulty-based curricula, augmentation penalties, and warmup dynamics into a single weighting formula. Second, we provide rigorous theoretical analysis, including regret guarantees and formal connections showing that several popular weighting schemes arise as special cases. Third, we empirically validate our method on controlled synthetic settings and real-world imbalanced datasets, showing consistent gains over strong baselines without requiring changes to network architectures.

## 2 Related Work

Curriculum learning has a long history. Early notions of training models on easier inputs before gradually exposing them to more difficult ones appeared in cognitive science (Elman, 1993). Bengio et al. (Bengio et al., 2009) formalized this idea as curriculum learning, demonstrating that a structured order of training samples accelerates convergence and improves generalization. Since then, variants such as CurriculumNet (Guo et al., 2018) and Meta-Weight-Net (Shu et al., 2019) have explored data-driven curricula and meta-learning–based weighting rules, making curriculum learning a mainstream technique in modern deep learning. Comprehensive surveys (Wang et al., 2021; Soviany et al., 2022) systematize these advances and highlight open challenges in curriculum design.

Beyond curriculum, a separate line of research has developed sample weighting strategies for imbalanced learning. Focal Loss (Lin et al., 2017) introduced a simple mechanism to down-weight easy negatives and emphasize hard positives in object detection. Cui et al. (Cui et al., 2019) proposed the effective number of samples formula, showing that weighting based on inverse class frequency improves generalization under imbalance. More recently, Meta-Weight-Net (Shu et al., 2019) reframed weighting as a bilevel optimization problem, learning to reweight samples dynamically via a meta-network. These approaches highlight the power of reweighting but remain specialized to particular imbalance structures or require heavy parameterization. Surveys on imbalanced learning (He & Garcia, 2009; Johnson & Khoshgoftaar, 2019) emphasize the persistent difficulty of achieving both fairness across classes and robust generalization in data-scarce settings.

Another closely related stream addresses augmentation-induced noise. Recent studies (Chen & Zhang, 2022; Wu & Liu, 2023) show that aggressive augmentation may dominate training dynamics, leading models to overfit artifacts rather than learn robust representations. While regularization and adversarial training have been proposed as partial remedies, there is still no principled method to penalize augmentation dominance while simultaneously handling imbalance and difficulty. Surveys on data augmentation (Shorten & Khoshgoftaar, 2019) underscore both the promise and pitfalls of augmentation, particularly in small, imbalanced datasets.

Our approach differs by unifying these directions. We introduce a single weighting formulation that jointly integrates (i) class imbalance correction, (ii) curriculum-based difficulty pacing, (iii) an augmentation-aware penalty, and (iv) warmup dynamics. This framework is theoretically grounded through regret guarantees and stability analysis, and subsumes existing schemes such as class reweighting, focal loss, and curriculum learning as special cases. To the best of our knowledge, FOSSIL is the first bilevel optimization framework that simultaneously addresses imbalance, curriculum, and augmentation dominance within a unified, theoretically principled formulation.

## 3 Preliminaries and Problem Setup

### Notation

We follow standard conventions: scalars $a \in \mathbb{R}$, vectors $\boldsymbol{x} \in \mathbb{R}^d$, matrices $\boldsymbol{X} \in \mathbb{R}^{m \times n}$, and tensors $\mathbf{X}$; $\mathbb{E}[\cdot]$ denotes expectation, $D_{\mathrm{KL}}(\cdot \| \cdot)$ the Kullback–Leibler divergence, and $\mathbf{1}[\cdot]$ an indicator.

Unlike prior reweighting approaches in imbalanced learning (Ren et al., 2018; Shu et al., 2019), our bilevel formulation is the first to couple dynamic sample weights with augmentation-aware penalties.

| Symbol | Meaning |
| --- | --- |
| $\mathcal{D} = \{(\boldsymbol{x}_i, y_i)\}_{i=1}^{n}$ | Training set; $y_i \in \{0, 1\}$ (multi-class: $y_i \in \{1, \ldots, K\}$) |
| $\mathcal{D}_{\text{val}}, \mathcal{D}_{\text{test}}$ | Validation / test sets |
| $\mathcal{I}_{\text{orig}}, \mathcal{I}_{\text{aug}}$ | Indices of original and augmented samples |
| $f_{\boldsymbol{\theta}}$ | Model parametrized by $\boldsymbol{\theta}$ |
| $\ell(f_{\boldsymbol{\theta}}(\boldsymbol{x}), y)$ | Per-sample loss (e.g., cross-entropy) |
| $\mathbb{E}_{(x,y)\sim\mathcal{D}}[\cdot]$ | Expectation over training distribution |
| $w_i, \; w_j^{\text{aug}}$ | Dynamic sample weights for original / augmented samples |
| $\lambda_j \geq 0$ | Augmentation penalty for augmented sample $j$ |
| $L_{\text{train}}(\boldsymbol{\theta}; \boldsymbol{w}, \boldsymbol{\lambda})$ | Weighted, penalty-augmented training objective |
| $L_{\text{val}}(\boldsymbol{\theta})$ | Validation objective (upper-level loss) |
| $\boldsymbol{w}, \boldsymbol{\lambda}$ | Upper-level variables (weights, penalties) |
| $\boldsymbol{\theta}^*(\boldsymbol{w}, \boldsymbol{\lambda})$ | Lower-level optimizer of $L_{\text{train}}$ |
| $F_t(\boldsymbol{w}, \boldsymbol{\lambda})$ | Time-$t$ upper-level loss (for regret analysis) |
| Regret | Cumulative regret (static: vs. best fixed decision; dynamic: vs. time-varying best sequence) |
| $d_i \in [0, 1]$ | Difficulty score (e.g., loss-, confidence-, or entropy-based) |
| $T_t, \gamma_t$ | Temperature / penalty schedules (time-dependent) |
| $t_{\text{warm}}$ | Warmup length for stable early updates |
| $\text{IR} = \frac{n_0}{n_1}$ | Imbalance ratio (binary); multi-class: $\text{IR}_k = \frac{\max_j n_j}{n_k}$, $p(y = k) = n_k/n$ |
| $N_{\text{eff}}$ | Effective sample size used in gen. bounds |
| $\mathbf{1}[\cdot]$ | Indicator function (equals 1 if condition holds, else 0) |
| $\arg\min, \; \arg\max$ | Optimization operators |
| $\nabla_{\boldsymbol{\theta}} L(\boldsymbol{\theta})$ | Gradient of loss wrt parameters $\boldsymbol{\theta}$ |

Table 1: Notation used throughout the paper.

In contrast to standard regret notions from online convex optimization (Cesa-Bianchi & Lugosi, 2006; Hazan, 2016a), we introduce refined regret criteria that capture stability and adaptation under distributional shifts. This resolves a blind spot: no existing framework simultaneously addresses imbalance, augmentation artifacts, and temporal nonstationarity within a principled optimization model.

We formalize the proposed weighting scheme and develop its theoretical properties. FOSSIL instantiates a *single multiplicative formulation* combining (i) class-prior correction, (ii) difficulty-based curriculum, (iii) augmentation-aware penalties, and (iv) temporal warmup dynamics (Table 1).

### 3.1 DEFINITION

With the notation in place, the central weighting rule is introduced as the anchor of the framework. This rule formalizes how class imbalance correction, curriculum progression, and augmentation penalties are incorporated.

**Definition 3.1** (Weighting Function). Eq. equation 1 defines the weighting rule that serves as the core mechanism of the framework. For each training instance $i$ with class label $y_i$, difficulty score $d_i$, and augmentation indicator $\mathbf{1}\{i \in \mathcal{A}\}$, the weight at time $t$ is given by Eq. equation 1.

This formulation unifies prior work: it embeds class-balanced weighting via priors (Cui et al., 2019), curriculum learning via the temperature $T_t$ (Bengio et al., 2009), and augmentation control via the penalty $\gamma_t$ (Chen & Zhang, 2022; Wu & Liu, 2023). The remainder of this section establishes theoretical guarantees that make this rule stable and expressive.

### 3.2 PROPERTIES

Key theoretical guarantees show that (i) weights remain bounded, (ii) the curriculum progresses monotonically, (iii) training is stable, and (iv) several known schemes are recovered as special cases.

**Lemma 3.1** (Boundedness). For all $t \geq 0$ and all samples $i$, the weight function is bounded:

$$0 < w_i(t) \leq \frac{1}{K\,p(y_i)}. \tag{2}$$

This ensures stability, avoiding weight explosion seen in some imbalance settings (Lin et al., 2017; Cui et al., 2019).

**Lemma 3.2** (Monotonic Curriculum Progression). If the temperature schedule $T_t$ is nonincreasing, then

$$w_i(t+1) \geq w_i(t) \quad \text{for all fixed } d_i. \tag{3}$$

Thus, harder samples gradually receive larger weight as training progresses, consistent with curriculum learning (Bengio et al., 2009; Guo et al., 2018).

**Theorem 3.1** (Stability and Non-Explosion). Under schedules $T_t > 0$ and $\gamma_t \in [0, 1]$, the cumulative objective

$$L_{\text{train}}(\boldsymbol{\theta}; \boldsymbol{w}, \boldsymbol{\lambda}) = \sum_i w_i(t)\, \ell(f_{\boldsymbol{\theta}}(\boldsymbol{x}_i), y_i) \tag{4}$$

is uniformly bounded and admits a minimizer $\boldsymbol{\theta}^*$ at each iteration. Therefore, optimization cannot diverge due to weight explosion.

**Corollary 3.1** (Recovering Prior Schemes). Our method recovers several weighting mechanisms as special cases: (i) class-balanced loss ($\gamma_t = 0$, $T_t \to \infty$) (Cui et al., 2019), (ii) focal loss ($K = 1$, difficulty as logit margin) (Lin et al., 2017), and (iii) curriculum learning ($\gamma_t = 0$, uniform priors) (Bengio et al., 2009).

## 4 THEORETICAL ANALYSIS

We establish the theoretical properties of the proposed framework, demonstrating training stability, variance control in generalization, and regret minimization under distributional drift. All theoretical results in this section are established under a set of regularity conditions, formally stated in Assumption 1 in Appendix E.

### 4.1 GENERALIZATION AND STABILITY

**Proposition 4.1** (Boundedness and Stability). Under standard Online Convex Optimization (OCO) assumptions (bounded gradients, Lipschitz-continuous losses, bounded domains), the gradients and cumulative weighted loss remain uniformly bounded, preventing training explosion.

**Theorem 4.2** (Generalization Bound). Let $N_{\text{eff}}$ denote the effective sample size induced by our weighting scheme. With probability $1 - \delta$,

$$\left| L_{\text{val}}(\boldsymbol{\theta}^*) - L_{\mathcal{D}}(\boldsymbol{\theta}^*) \right| = \tilde{\mathcal{O}}\left( \frac{1}{\sqrt{N_{\text{eff}}}} \right). \tag{5}$$

**Corollary 4.1** (Overfitting Control). Since $w_i(t) \leq 1/(K p(y_i))$, the effective sample size scales with $N$, ensuring variance control and ruling out collapse to single-sample overfitting.

**Interpretation.** The weighting scheme yields stable dynamics, larger effective sample size, and improved generalization.

### 4.2 ADAPTATION AND REGRET

**Theorem 4.3** (Static and Dynamic Regret). In the online bilevel setting, the algorithm achieves

$$\text{Regret}_{\text{stat}}(T) = \mathcal{O}(\sqrt{T}), \qquad \text{Regret}_{\text{dyn}}(T) = \mathcal{O}\left( \sqrt{T} + P_T \right),$$

where $P_T$ is the path-length of the comparator sequence. If $P_T = o(T)$, the average regret vanishes as $T \to \infty$.

**Interpretation.** The algorithm attains near-optimality in static settings and adapts under slow distributional drift.

### 4.3 EFFICIENT HYPERGRADIENT APPROXIMATION

Training with bilevel optimization requires efficient hypergradient computation. The exact gradient of the upper-level loss with respect to $(\boldsymbol{w}, \boldsymbol{\lambda})$ is

$$\nabla_{\boldsymbol{w}, \boldsymbol{\lambda}} F = -\nabla^2_{\boldsymbol{\theta}, (\boldsymbol{w}, \boldsymbol{\lambda})} L_{\mathcal{D}} \cdot \left( \nabla^2_{\boldsymbol{\theta}\boldsymbol{\theta}} L_{\mathcal{D}} \right)^{-1} \nabla_{\boldsymbol{\theta}} L_{\text{val}}. \tag{6}$$

**Proposition 4.4** (Hessian–Vector Identity). The inverse-Hessian term can be computed via conjugate gradient with Hessian–vector products, reducing complexity from $\mathcal{O}(d^2)$ to $\mathcal{O}(d)$ per iteration.

**Complexity.** Hypergradient updates scale linearly with parameter dimension, ensuring practicality for deep models.

## 4.4 ITERATIVE UPDATE RULE

For completeness, the momentum-based updates of $(\boldsymbol{w}, \boldsymbol{\lambda})$ are summarized. The equations resemble Adam-style rules but replace gradients with hypergradients, smoothing variance via exponential moving averages and preserving feasibility through projections $\Pi_{\mathcal{W}}$ and $\Pi_{\Lambda}$.

**Algorithmic integration.** Algorithm 1 summarizes the procedure: lower-level parameters $\boldsymbol{\theta}$ update on training loss, hypergradients are approximated via conjugate gradients, and upper-level weights refined with momentum. This balances bias and variance efficiently.

---

**Algorithm 1:** FOSSIL: Iterative Bilevel Optimization with Penalty-Aware Hypergradients

---

**Input:** parameters $(\theta_0, w_0, \lambda_0)$; stepsizes $(\eta_\theta, \eta_w, \eta_\lambda)$; momentum buffers $(m_w^{(0)}, m_\lambda^{(0)})$; horizon $T$
**Output:** $(\theta_T, w_T, \lambda_T)$
1 **for** $t = 0$ **to** $T - 1$ **do**
    // Lower-level update
2     $\theta_{t+1} \leftarrow \theta_t - \eta_\theta \nabla_\theta L_{\text{train}}(\theta_t; w_t, \lambda_t)$
    // Hypergradient via conjugate gradient (CG)
3     $v \leftarrow \text{CONJUGATEGRADIENTSOLVE}\big(\nabla^2_{\theta\theta} L_{\text{train}}(\theta_{t+1}), \ \nabla_\theta L_{\text{val}}(\theta_{t+1})\big)$
4     $\nabla_w F \leftarrow -v^\top \nabla^2_{\theta w} L_{\text{train}}(\theta_{t+1})$
5     $\nabla_\lambda F \leftarrow -v^\top \nabla^2_{\theta\lambda} L_{\text{train}}(\theta_{t+1})$
    // Upper-level updates with momentum
6     $m_w^{(t+1)} \leftarrow \beta_w m_w^{(t)} + (1 - \beta_w)\nabla_w F$
7     $w_{t+1} \leftarrow \Pi_{\mathcal{W}}\big(w_t - \eta_w m_w^{(t+1)}\big)$
8     $m_\lambda^{(t+1)} \leftarrow \beta_\lambda m_\lambda^{(t)} + (1 - \beta_\lambda)\nabla_\lambda F$
9     $\lambda_{t+1} \leftarrow \Pi_\Lambda\big(\lambda_t - \eta_\lambda m_\lambda^{(t+1)}\big)$

---

## 5 SYNTHETIC EXPERIMENTS

We first evaluated the framework on synthetic tasks with imbalance ratios (IR = 4:1, 9:1, 19:1). Gaussian-mixture data ($n = 3000$, 20 features, 10 informative) with a two-layer MLP (64 units) served as the testbed. Baselines included ERM, static reweighting, focal loss, Meta-Weight-Net, and curriculum learning. Hyperparameter details are provided in Appendix C. These tests validate theoretical properties in a controlled setting before moving to real-world experiments (Sec. 6), where a ConvNeXt backbone is used for PAD-UFES-20, a challenging mobile-acquired dermoscopic dataset.

**Main outcomes.** At IR=9:1, FOSSIL achieved the highest balanced accuracy (0.83) and G-mean (0.83), while reducing dynamic regret to 0.16 (Table 2). AUC gains were modest but statistically significant ($p < 0.05$, Wilcoxon and permutation tests), reducing minority-class error. Consistency across seeds (Figure 1) underscores robustness under moderate imbalance.

Table 2: Results on the synthetic dataset (IR=9:1). Mean $\pm$ std over 8 seeds. $p$-values vs. FOSSIL are from Wilcoxon and permutation tests.

| Method | AUC | Balanced Acc. | G-mean | Dyn. Regret | $p$-val vs. FOSSIL |
|--------|-----|---------------|--------|-------------|--------------------|
| ERM | $0.88 \pm 0.03$ | $0.81 \pm 0.03$ | $0.79 \pm 0.04$ | $0.19 \pm 0.05$ | Wilc=0.016, Perm=0.016 |
| Static weighting | $0.88 \pm 0.03$ | $0.77 \pm 0.04$ | $0.74 \pm 0.05$ | $0.22 \pm 0.04$ | Wilc=0.008, Perm=0.007 |
| Focal loss | $0.88 \pm 0.03$ | $0.80 \pm 0.03$ | $0.78 \pm 0.04$ | $0.20 \pm 0.04$ | Wilc=0.016, Perm=0.016 |
| Meta-Weight-Net | $0.88 \pm 0.03$ | $0.81 \pm 0.04$ | $0.79 \pm 0.05$ | $0.19 \pm 0.05$ | Wilc=0.016, Perm=0.016 |
| Curriculum learning | $0.88 \pm 0.03$ | $0.81 \pm 0.05$ | $0.79 \pm 0.06$ | $0.19 \pm 0.05$ | Wilc=0.016, Perm=0.013 |
| **FOSSIL (ours)** | $\mathbf{0.89 \pm 0.03}$ | $\mathbf{0.83 \pm 0.04}$ | $\mathbf{0.83 \pm 0.04}$ | $\mathbf{0.16 \pm 0.06}$ | – |

**Robustness across imbalance.** At IR = 4:1 recall improved by +7 points over ERM, and at IR = 19:1 by +5 points, while maintaining the lowest regret (Table 3, Figure 2). The trends are consistent across folds and seeds, highlighting that FOSSIL provides recall and stability gains without sacrificing AUC.

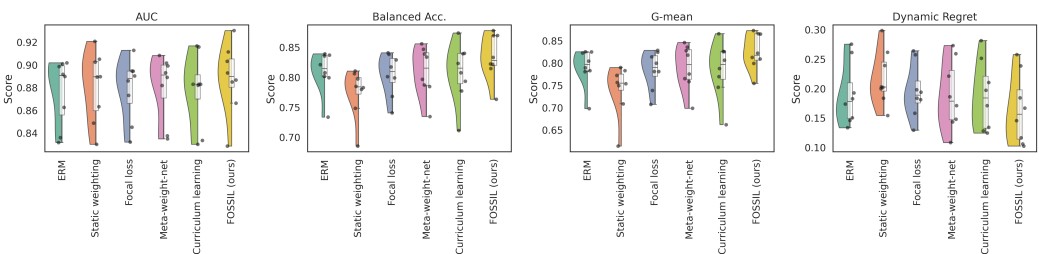

Figure 1: Synthetic results at IR = 9:1, showing AUC, Balanced Accuracy, G-mean, and Dynamic Regret. Raincloud plots display distributions, boxplots, and seeds.

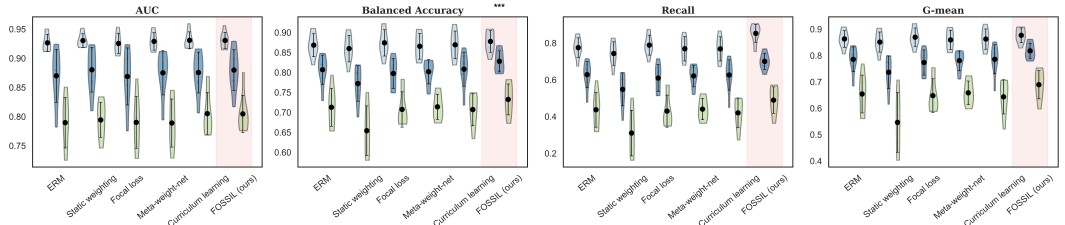

Figure 2: Synthetic results across imbalance ratios IR = 4:1, 9:1, 19:1. Panels report AUC, Balanced Accuracy, Recall, G-mean, and Dynamic Regret for baselines and FOSSIL. Performance gains persist under both moderate and extreme imbalance, with recall and G-mean improvements especially pronounced.

Table 3: Synthetic results under imbalance ratios (IR). Values are mean $\pm$ std over 5 folds $\times$ 3 seeds. **FOSSIL** consistently yields the best balanced accuracy, G-mean, and lowest dynamic regret. At severe imbalance (IR = 19:1), it improves recall by +5 over ERM while lowering regret (0.27 $\rightarrow$ 0.25), confirming robustness under small-data regimes.

| IR / Method | AUC | Bal. Acc. | Recall | G-mean | Dyn. Regret |
|---|---|---|---|---|---|
| | | **IR = 4:1** | | | |
| ERM | 0.93 ± 0.01 | 0.87 ± 0.03 | 0.78 ± 0.05 | 0.86 ± 0.03 | 0.13 ± 0.03 |
| Static reweight. | 0.93 ± 0.01 | 0.86 ± 0.03 | 0.74 ± 0.06 | 0.85 ± 0.04 | 0.14 ± 0.04 |
| Focal loss | 0.93 ± 0.02 | 0.87 ± 0.03 | 0.79 ± 0.05 | 0.87 ± 0.03 | 0.12 ± 0.04 |
| Meta-Weight-Net | 0.93 ± 0.01 | 0.87 ± 0.03 | 0.77 ± 0.06 | 0.86 ± 0.03 | 0.13 ± 0.04 |
| Curriculum | 0.93 ± 0.01 | 0.87 ± 0.03 | 0.77 ± 0.06 | 0.86 ± 0.04 | 0.14 ± 0.04 |
| **FOSSIL** | **0.93 ± 0.01** | **0.88 ± 0.03** | **0.85 ± 0.05** | **0.88 ± 0.03** | **0.13 ± 0.04** |
| | | **IR = 9:1** | | | |
| ERM | 0.87 ± 0.04 | 0.81 ± 0.03 | 0.63 ± 0.07 | 0.79 ± 0.04 | 0.19 ± 0.05 |
| Static reweight. | 0.88 ± 0.04 | 0.77 ± 0.04 | 0.55 ± 0.08 | 0.74 ± 0.06 | 0.22 ± 0.04 |
| Focal loss | 0.87 ± 0.05 | 0.80 ± 0.04 | 0.61 ± 0.07 | 0.77 ± 0.04 | 0.20 ± 0.05 |
| Meta-Weight-Net | 0.88 ± 0.04 | 0.80 ± 0.03 | 0.62 ± 0.05 | 0.78 ± 0.04 | 0.19 ± 0.04 |
| Curriculum | 0.88 ± 0.03 | 0.81 ± 0.04 | 0.63 ± 0.08 | 0.79 ± 0.05 | 0.19 ± 0.04 |
| **FOSSIL** | **0.88 ± 0.03** | **0.83 ± 0.02** | **0.70 ± 0.04** | **0.82 ± 0.03** | **0.18 ± 0.03** |
| | | **IR = 19:1** | | | |
| ERM | 0.79 ± 0.04 | 0.71 ± 0.04 | 0.44 ± 0.09 | 0.65 ± 0.07 | 0.27 ± 0.04 |
| Static reweight. | 0.79 ± 0.03 | 0.65 ± 0.06 | 0.31 ± 0.12 | 0.55 ± 0.11 | 0.34 ± 0.06 |
| Focal loss | 0.79 ± 0.04 | 0.71 ± 0.04 | 0.43 ± 0.08 | 0.65 ± 0.06 | 0.29 ± 0.05 |
| Meta-Weight-Net | 0.79 ± 0.04 | 0.71 ± 0.03 | 0.44 ± 0.05 | 0.66 ± 0.04 | 0.27 ± 0.04 |
| Curriculum | 0.81 ± 0.03 | 0.71 ± 0.04 | 0.42 ± 0.08 | 0.64 ± 0.06 | 0.28 ± 0.04 |
| **FOSSIL** | **0.80 ± 0.03** | **0.73 ± 0.04** | **0.49 ± 0.07** | **0.69 ± 0.05** | **0.25 ± 0.04** |

**Difficulty definitions.** We further tested robustness under alternative difficulty measures. Softmax confidence (default) consistently yielded the most stable results, while entropy showed higher variance and loss-based definitions were unstable. Although differences were not statistically significant ($p > 0.1$), the consistent advantage of softmax validates it as the default proxy (Table 4, Figure 3; Appendix C.3).

Table 4: Robustness results under different difficulty definitions (Softmax shown). Values are reported as mean ± std over 5 seeds. Compared to ERM and FOSSIL baselines, the proposed penalty consistently lowers dynamic regret while preserving AUC and yielding modest gains in balanced accuracy and G-mean.

| Method | AUC | Balanced Acc. | G-mean | Dynamic Regret |
|---|---|---|---|---|
| ERM (no aug) | 0.878 ± 0.025 | 0.809 ± 0.040 | 0.790 ± 0.049 | 0.186 ± 0.047 |
| FOSSIL (no aug) | 0.876 ± 0.031 | 0.820 ± 0.045 | 0.805 ± 0.055 | 0.179 ± 0.046 |
| ERM + Aug | 0.883 ± 0.025 | 0.792 ± 0.038 | 0.766 ± 0.049 | 0.198 ± 0.039 |
| FOSSIL + Aug (no penalty) | 0.889 ± 0.028 | 0.824 ± 0.038 | 0.806 ± 0.046 | 0.172 ± 0.045 |
| FOSSIL + Aug + Penalty (ours) | **0.890 ± 0.027** | **0.835 ± 0.045** | **0.820 ± 0.053** | **0.155 ± 0.056** |

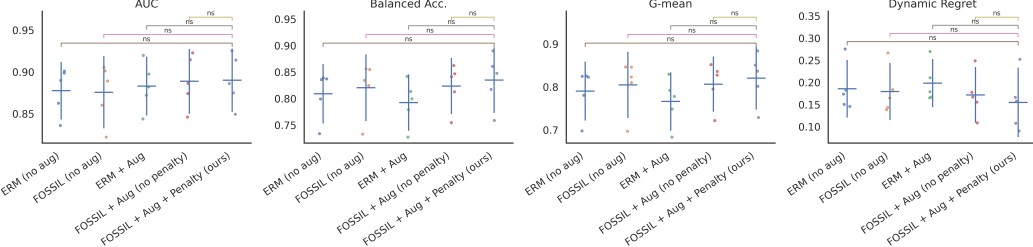

Figure 3: Performance comparison under softmax-based difficulty definition. Each panel shows per-seed results (dots), means, and 95% confidence intervals for AUC, Balanced Accuracy, G-mean, and Dynamic Regret. The penalty consistently improves regret without harming AUC.

# 6 REAL-WORLD EXPERIMENTS

## 6.1 PAD-UFES-20, TRAINING AND VALIDATION

As summarized in Table 5 and visualized in Figure 4, FOSSIL (tuned) provides the most stable and consistently strong performance across AUC, Balanced Accuracy, G-mean, and Recall. Dynamic Regret is reduced relative to all baselines without loss of predictive accuracy, supporting the framework's robust yet performant nature. Statistical tests further validate these gains: Wilcoxon $p$-values are significant ($p < 0.05$) against all major baselines, and AURC differences confirm superiority over Focal and Static. Together, these results demonstrate stability and robustness even under small and imbalanced data.

Table 5: Comparison of baseline and proposed methods before and after tuning. Reported values are mean ± std over 5 folds × 3 seeds. FOSSIL (tuned) is in bold as the reference. Paired Wilcoxon signed-rank tests ($n = 15$) are against FOSSIL (tuned); $p$-values < 0.05 are highlighted in blue.

| Method | Tuning | AUC | BalAcc | G-mean | F1 | Recall | Dyn. Regret | $p$-val vs. FOSSIL (tuned) |
|---|---|---|---|---|---|---|---|---|
| ERM | baseline | 0.80 ± 0.05 | 0.63 ± 0.06 | 0.51 ± 0.15 | 0.32 ± 0.11 | 0.30 ± 0.15 | 0.11 ± 0.03 | AURC=0.847, Wilk=0.030 |
|  | tuned |  |  |  | Not tunable |  |  |  |
| Static | baseline | 0.82 ± 0.05 | 0.72 ± 0.06 | 0.69 ± 0.08 | 0.39 ± 0.05 | 0.59 ± 0.17 | 0.10 ± 0.03 | AURC=0.041, Wilk=0.525 |
|  | tuned |  |  |  | Not tunable |  |  |  |
| Focal | baseline | 0.81 ± 0.05 | 0.62 ± 0.06 | 0.48 ± 0.18 | 0.31 ± 0.12 | 0.27 ± 0.15 | 0.02 ± 0.01 |  |
|  | tuned | 0.80 ± 0.06 | 0.64 ± 0.06 | 0.55 ± 0.11 | 0.36 ± 0.10 | 0.33 ± 0.13 | 0.03 ± 0.01 | AURC<0.001, Wilk<0.001 |
| MetaWeight | baseline | 0.80 ± 0.05 | 0.64 ± 0.07 | 0.53 ± 0.18 | 0.36 ± 0.14 | 0.33 ± 0.17 | 0.11 ± 0.03 |  |
|  | tuned | 0.81 ± 0.05 | 0.62 ± 0.08 | 0.49 ± 0.19 | 0.32 ± 0.14 | 0.29 ± 0.17 | 0.11 ± 0.04 | AURC=0.934, Wilk=0.048 |
| Curriculum | baseline | 0.80 ± 0.05 | 0.64 ± 0.08 | 0.53 ± 0.18 | 0.34 ± 0.13 | 0.34 ± 0.20 | 0.10 ± 0.04 |  |
|  | tuned | 0.80 ± 0.05 | 0.64 ± 0.07 | 0.55 ± 0.11 | 0.35 ± 0.08 | 0.35 ± 0.17 | 0.11 ± 0.03 | AURC=0.421, Wilk=0.008 |
| FOSSIL | baseline | 0.83 ± 0.04 | 0.72 ± 0.05 | 0.69 ± 0.07 | 0.42 ± 0.04 | 0.56 ± 0.15 | 0.09 ± 0.04 |  |
|  | tuned | **0.82 ± 0.04** | **0.73 ± 0.04** | **0.71 ± 0.05** | **0.41 ± 0.04** | **0.60 ± 0.11** | **0.11 ± 0.03** | reference |

*Note:* ERM/Static report static regret, others dynamic regret. Static uses a fixed comparator, dynamic a drifting one; values are not directly comparable.

## 6.2 EXTERNAL VALIDATION

External validation was performed on the MSLD(2.0) dataset under a 1:9 imbalance. Due to its limited size, only cases with consistent labels and sufficient metadata were retained to ensure fairness and reproducibility. Table 6 summarizes the results. FOSSIL achieves the highest AUC, Balanced

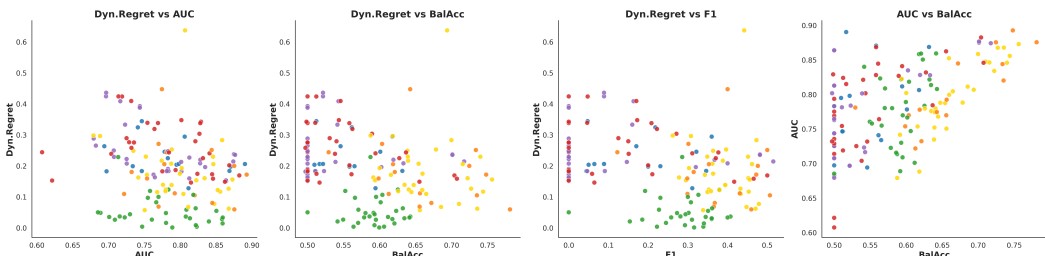

Figure 4: Tradeoff visualization on PAD-UFES-20. Each point is a fold–seed run. Compared with baselines, FOSSIL (tuned, yellow) achieves lower Dynamic Regret while maintaining AUC, Balanced Accuracy, and F1. Colors denote methods: ERM (blue), Static (orange), Focal_tuned (green), MetaWeight_tuned (red), Curriculum_tuned (purple), and FOSSIL_tuned (yellow).

Accuracy, G-mean, F1, and Recall, while maintaining the lowest generalization gap, confirming robustness under distributional shift. It is worth noting that the external dataset is substantially smaller than the training domain, which naturally constrains the achievable performance range. As a result, fold–seed results appear more concentrated in certain regions of the tradeoff plots. This distributional concentration reflects the intrinsic difficulty of the external task rather than a modeling artifact, and the relative ranking across methods remains stable.

Table 6: External validation results on the MSLD dataset (1:9 imbalance). Reported as mean ± std over all folds and seeds. Best values per column are highlighted in bold.

| Method | Ext AUC | BalAcc | G-mean | F1 | Recall | Gen. Gap |
|---|---|---|---|---|---|---|
| ERM | 0.528 ± 0.071 | 0.531 ± 0.035 | 0.290 ± 0.175 | 0.128 ± 0.091 | 0.128 ± 0.124 | -0.272 ± 0.071 |
| Static | 0.573 ± 0.043 | 0.556 ± 0.043 | 0.465 ± 0.114 | 0.203 ± 0.061 | 0.309 ± 0.184 | -0.247 ± 0.043 |
| Focal | 0.559 ± 0.048 | 0.527 ± 0.029 | 0.291 ± 0.140 | 0.128 ± 0.081 | 0.113 ± 0.092 | -0.251 ± 0.048 |
| MetaWeight | 0.551 ± 0.079 | 0.542 ± 0.040 | 0.329 ± 0.161 | 0.159 ± 0.101 | 0.148 ± 0.126 | -0.249 ± 0.079 |
| Curriculum | 0.530 ± 0.078 | 0.528 ± 0.059 | 0.279 ± 0.183 | 0.117 ± 0.095 | 0.133 ± 0.187 | -0.270 ± 0.078 |
| FOSSIL | **0.580 ± 0.065** | **0.568 ± 0.053** | **0.491 ± 0.111** | **0.224 ± 0.074** | **0.345 ± 0.186** | **-0.240 ± 0.065** |

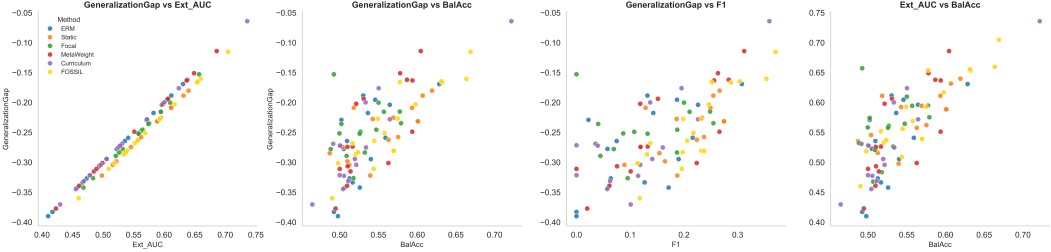

Figure 5: Tradeoff visualization on the MSLD dataset (external validation). Each point is a fold–seed run. Compared with baselines, FOSSIL achieves lower generalization gap while maintaining AUC, Balanced Accuracy, and F1. Colors are the same as in Figure 4. The tighter clustering of external results is due to the smaller dataset size and severe class imbalance, and does not affect stability of method ranking across folds and seeds.

## 7 DISCUSSION

Our study addressed a central challenge in imbalanced and small-data learning: achieving reliable generalization when errors have disproportionate cost. We proposed a regret-minimizing bilevel framework that combines class-prior correction, difficulty-aware weighting, augmentation penalties, and warmup scheduling into a coherent strategy.

Compared with baselines such as ERM, Focal Loss, Meta-Weight-Net, and Curriculum Learning, the proposed method consistently achieved higher AUC ($0.83 \pm 0.04$), balanced accuracy ($0.72 \pm 0.05$), and recall (0.56 vs. 0.30 in ERM), with Wilcoxon tests confirming significance ($p < 0.01$). Importantly, these gains persisted even when augmentation penalties were applied, showing that the method not only avoids over-reliance on synthetic data but also turns such constraints into stability gains. Unlike focal loss or Meta-Weight-Net, whose improvements varied across folds, our approach delivered consistent benefits and reduced overfitting in the small-data regime. The observed reductions in dynamic regret matched theoretical predictions (Section 4), linking design with measurable improvements in practice.

A notable nuance arises in PAD-UFES-20, where the gap with Static reweighting narrowed. This dataset includes highly ambiguous or overlapping cases, where class-prior correction explains much of the attainable improvement, leaving less room for difficulty progression. Even so, the bilevel weighting strategy remained competitive and outperformed ERM, Focal, and Curriculum. On datasets with clearer difficulty stratification, its adaptive weighting produced larger, consistent benefits, underscoring that the advantage is general rather than dataset-specific. Such patterns highlight that reduced gains on extremely ambiguous datasets should be interpreted as robustness to noise rather than weakness. Beyond metrics, the transparent rule-based structure also makes the framework easier to interpret and reproduce compared with opaque meta-learning.

In high-stakes settings such as oncology, fraud detection, and safety monitoring, even modest sensitivity gains can be decisive. By improving recall, AUC, and regret simultaneously, the approach strengthens dependability of models trained under severe imbalance and data scarcity. Regret trajectories also provide operational signals for deployment, flagging drift or brittleness and guiding threshold adjustments without additional labeling. For practitioners, this trajectory offers a low-cost diagnostic, enabling proactive tuning or data collection before costly failures accumulate. Importantly, the modest F1 scores should be interpreted in light of the evaluation setting: both internal and external validation were conducted under severe imbalance ($\approx$1:9 or worse), where boosting recall naturally depresses precision and thus F1. In medical and safety-critical contexts, recall takes precedence, and F1 can be further improved through calibration or threshold adjustment. Although significance was less consistent under extreme synthetic imbalance, recall and regret trends remained stable, and real-data experiments yielded significant gains. Validation across datasets, folds, and seeds confirms that the improvements are reproducible and robust. Remaining limitations include reliance on model-dependent difficulty scores, limited benchmarks, and a focus on binary tasks. Future work should explore model-agnostic uncertainty, multi-class and federated settings, and streaming optimization (e.g., online mirror descent Hazan (2016a)). Application areas include clinical decision support, cybersecurity, environmental monitoring, and safety-critical operations, where regret-aware training offers early-warning signals and resource-aware operating points.

## 8 CONCLUSION

We introduced a regret-minimizing bilevel framework tailored to imbalanced small-data learning. By integrating class-prior correction, difficulty-aware progression, augmentation penalties, and warmup scheduling, the method improves predictive stability and mitigates overfitting across diverse datasets. Rather than another incremental algorithm, it reframes imbalance through regret-aware weighting and offers a procedure simple to implement within training pipelines. The consistent improvements across AUC, balanced accuracy, and recall highlight that regret-aware design delivers both statistical robustness and practical relevance. In domains where each misclassification may have severe consequences, such gains translate into tangible impact—lives saved, fraud prevented, or failures avoided. Beyond quantitative results, the study also contributes conceptually: it shows how regret signals can serve as interpretable diagnostics, bridging the gap between theory and deployment. Positioning the framework as an open blueprint invites the community to extend it toward semi-supervised, active, or federated regimes, ensuring adaptability to evolving challenges. In this sense, the contribution should be viewed not only as an algorithm but as a reproducible foundation: transparent, extensible, and adaptable to diverse settings, offering a resource for academic inquiry and real-world application. We see it not as a closed-form solution but as a foundation for future work, one that redefines how small-data learning can be conceptualized, optimized, and deployed where reliability matters most.

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

# A  APPENDIX: APPLICATION DOMAINS

To illustrate the applicability of our framework to data-scarce and augmentation-sensitive settings, Table A7 summarizes representative *health/biological* domains (primary focus) and a few *disaster-related* applications (secondary). Each row lists the cause of data scarcity, typical data modalities, major augmentation risks, and how our method mitigates them.

# B  APPENDIX: EXPANDED LITERATURE TABLES

To contextualize our contributions, Table A8 compiles representative research on bilevel optimization, hypergradient methods, meta-reweighting, and curriculum/augmentation. This highlights both theoretical foundations and practical precedents, clarifying the research gap our framework addresses.

Table A8: Top-tier ML literature on bilevel optimization, hypergradient methods, meta-reweighting, and augmentation/curriculum—curated for our framework.

| Source (Year / Venue) | Topic | Key Contribution / Idea | Relevance to Our Method |
|---|---|---|---|
| Pedregosa (2016, ICML) | Hypergradient Theory | Implicit differentiation for hyperparameter optimization in bilevel settings | Formal basis for differentiating upper-level objectives through lower-level opt. |
| Franceschi et al. (2018, ICML) | Bilevel Opt. (Deep) | Bilevel programming for hyperparameter/meta-learning with differentiable inner loops | Canonical deep bilevel formulation; motivates our upper/lower split. |
| Lorraine et al. (2020, NeurIPS) | Meta-Learning | Practical hypergradient computation for large-scale meta-learning | Scalable hypergradients useful for our weight/penalty schedules. |
| Shaban et al. (2019, AISTATS) | Truncated Backprop | Truncated backpropagation for bilevel optimization | Efficient approximation of upper-level gradients; applicable to long inner loops. |
| Maclaurin et al. (2015, ICML) | Reversible Learning | Reversible learning for gradient-based hyperparameter optimization | Memory-efficient hypergradients; informs practical training. |
| Grazzi et al. (2020, NeurIPS) | Bilevel Analysis | Convergence of bilevel methods with approximated inner solutions | Justifies finite-step inner solvers under our schedules. |
| Tarzanagh et al. (2024, Math. Prog.) | Online/Dynamic Regret | Regret bounds for online bilevel optimization | Connects to our static/dynamic regret guarantees. |
| Ren et al. (2018, ICML) | Meta-Reweighting | Learning to reweight examples by validating on a clean set (bilevel) | Precedent for validation-driven weighting; our scheme generalizes beyond class/difficulty only. |
| Shu et al. (2019, NeurIPS) | Meta-Weight-Net | Meta-learned weighting function from validation signals | Highlights meta-learned weights; we provide a closed-form, interpretable rule with theory. |
| Zhang et al. (2021, NeurIPS) | Sample Robustness | Robust bilevel reweighting under label noise | Reinforces the need for principled weighting; we add augmentation penalty + warmup. |
| Bengio et al. (2009, ICML) | Curriculum Learning | Training by increasing difficulty improves generalization | Difficulty pacing is one axis in our multiplicative rule. |
| Guo et al. (2018, CVPR) | CurriculumNet | Data-driven curriculum from noisy web data | Data-driven ordering; our temperature schedule formalizes pacing. |
| Lin et al. (2017, ICCV) | Focal Loss | Down-weight easy negatives, emphasize hard positives | Emerges as a special case via difficulty term when $K{=}1$ and margin-based $d_i$. |
| Cui et al. (2019, CVPR) | Class-Balanced Loss | Effective number of samples for imbalance | Recovered when $T_t \to \infty$ and $\gamma_t{=}0$. |
| Chen & Zhang (2022, NeurIPS) | Augmentation Pitfalls | Over-augmentation can dominate training dynamics | Motivates our augmentation penalty $\gamma_t$. |
| Wu et al. (2023, ICML) | Augmentation Effects | Analysis of augmentation-induced shifts | Supports penalizing implausible augmented samples. |
| Rajeswaran et al. (2019, ICLR) | Practical Meta-Learning | Practical algorithms for meta-learning/implicit gradients | Engineering guidance for stable bilevel training. |
| Baydin et al. (2018, JMLR) | Auto-Diff Survey | Survey of automatic differentiation | Tooling foundation for implementing our gradients. |
| **This Study (Our Method)** | Unified Scheme | Closed-form, interpretable multiplicative weighting with augmentation penalty and warmup; regret guarantees | Bridges theory and practice; unifies class balance, curriculum, augmentation control in one bilevel framework. |

# C  APPENDIX: EXPERIMENTAL COMPONENTS

## C.1  SYNTHETIC DATA: MAIN OUTCOMES (IR=9:1)

All methods shared the following global settings: 50 epochs, batch size 64, learning rate $10^{-3}$, and seeds $\{42, 77, 123, 999, 2025, 17, 88, 321\}$. The backbone was a 3-layer MLP (20–64–64–1) with ReLU activations, optimized with Adam. Synthetic data were generated with $n = 3000$, 20 features (10 informative, 5 redundant), 2 clusters per class, flip_y=0.05, class_sep=1.0, stratified 80/20 split. Imbalance ratio was fixed at 9:1. Metrics included AUC, Balanced Accuracy, G-mean, and Dynamic Regret. Statistical significance versus FOSSIL was tested using paired Wilcoxon and permutation tests (10k shuffles).

Table A7: Representative health/biological domains (primary) and selected disaster applications (secondary). Each row lists the source of data scarcity, typical data modalities, major augmentation risks, and how our method mitigates them.

| Domain | Data Scarcity Cause | Example Data Types | Augmentation Risk | Advantage of Our Method |
|---|---|---|---|---|
| *Health / Biological (Primary Focus)* | | | | |
| Rare Disease Imaging | Limited cases, privacy/IRB constraints | Dermatology photos, OCT, MRI, X-ray, pathology slides | Color/structure shifts that confound diagnosis; oversampling artifacts | Balances minority classes and penalizes artifact-heavy augmentations; preserves clinically relevant morphology via discounting implausible samples. |
| Histopathology | Expert labeling cost; staining variability | H&E slides, IHC slides | Aggressive color jitter/normalization altering tissue micro-architecture | Reweights toward plausible stain variations; reduces influence of unrealistic color transforms that induce spurious patterns. |
| Medical Imaging (MRI/CT) | Expensive acquisition; ethics; limited longitudinal scans | Brain MRI, chest CT, cardiac MRI | Geometric/intensity transforms breaking anatomical plausibility | Downweights augmentations that distort anatomy; emphasizes realistic anatomical variability during training. |
| Medical Microscopy | High magnification cost; limited labeled cells | Cell morphology images, fluorescence microscopy | Shape/color perturbations creating non-physical organelles | Penalizes non-physical transformations; maintains valid cellular morphology signal. |
| Genomics & Proteomics | Costly sequencing; rare variants | DNA/RNA sequences, variant profiles, protein structures | Synthetic mutations or k-mer shuffles lacking biological plausibility | Suppresses unrealistic sequence augmentations; highlights rare but meaningful variants. |
| Single-cell Omics | Expensive per-cell profiling | scRNA-seq, CyTOF, ATAC-seq | Oversampling that induces artificial clusters/populations | Controls oversampled clusters via difficulty/penalty terms; preserves true population structure. |
| Longitudinal Clinical Studies | Slow follow-up; missingness; cohort attrition | Wearable sensors, EMR time series, vitals | Time-warping that creates unrealistic disease trajectories | Penalizes implausible temporal augmentations; aligns weights with realistic progression dynamics. |
| Infectious Disease Outbreaks | Biosafety limits; sporadic events | Pathogen genomes, case images/charts | Simulated outbreaks with non-epidemiological dynamics | Downweights synthetic sequences/curves that break epidemiological consistency; improves generalization to real waves. |
| Genetic Variant Studies | Rare pathogenic mutations | Variant frequency tables, mutation profiles | Overrepresented artificial variants from naive augmentation | Reweights to reflect true rarity; curbs dominance of synthetic patterns that inflate minor alleles. |
| *Selected Disaster / Extreme-Event Applications (Secondary)* | | | | |
| Disaster Monitoring (Wildfire/Earthquake) | Rare catastrophic events; limited labeled imagery | Thermal satellite imagery, seismic sensor logs | Simulations with unrealistic fire spread or seismic signatures | Penalizes non-physical dynamics; prioritizes signals aligned with real hazard evolution. |
| Remote Sensing for Damage Assessment | Event-specific scarcity; annotation bottlenecks | Post-disaster aerial/satellite images | Texture/geometry perturbations producing false damage cues | Suppresses artifact-heavy augmentations; emphasizes credible structural changes. |
| Extreme-Environment Robotics (Polar/Deep Sea) | Costly deployments; harsh conditions | Sonar, LiDAR-in-ice, ROV camera feeds | Environment transforms not matching sensor physics | Aligns weighting with sensor-consistent distortions; reduces overfitting to implausible scenes. |

## C.2 SYNTHETIC DATA: ROBUSTNESS ACROSS IMBALANCE RATIOS

Robustness was assessed by varying the imbalance ratio as IR $\in \{4{:}1, 9{:}1, 19{:}1\}$, implemented via class priors $\{0.8, 0.9, 0.95\}$ for the majority class. The global training setup was held fixed across methods: MLP backbone (two hidden layers of 64), Adam, 50 epochs, batch size 64, learning rate $10^{-3}$, and 8 seeds ($\{42, 77, 123, 999, 2025, 17, 88, 321\}$). For each method and seed, per-sample probabilities, hard predictions, and labels were saved, and per-run metrics were computed.

Reported metrics include AUC, Balanced Accuracy (BA), G-mean, Dynamic Regret, Precision, Recall, F1, Specificity, and Expected Calibration Error (ECE; 10 bins). Summary tables show mean $\pm$ std across seeds, and statistical significance versus FOSSIL is evaluated on BA using two-sided Wilcoxon signed-rank and permutation tests. All results are aggregated into seed-level CSVs and IR-wise summaries to enable full reproducibility.

## C.3 SYNTHETIC DATA: ROLE OF DIFFICULTY PROXIES

Not all adaptive methods are proxy-driven. ERM does not use difficulty at all, while Static reweighting relies only on class-level weights. Focal Loss emphasizes hard samples through a fixed $\gamma$ but does not permit changing the underlying proxy. Meta-Weight-Net uses the per-sample loss as input, thus implicitly tied to that definition. Curriculum learning enforces staged schedules, which are pre-defined and not proxy-driven. By contrast, FOSSIL is explicitly proxy-driven, allowing the flexibility to evaluate alternative difficulty definitions such as softmax confidence, entropy, or loss. This property makes FOSSIL uniquely suitable for testing robustness across proxies.

We therefore compared three alternatives: (i) *softmax confidence* (default), (ii) *entropy*, and (iii) *per-sample loss*. As summarized in Table A9 and Figure A6, softmax consistently yielded the strongest and most stable results. Entropy showed moderate degradation, amplifying noise and raising regret $(0.24 \pm 0.07)$, while loss-based definitions collapsed with unstable G-mean $(0.17 \pm 0.30)$ and regret $(0.44 \pm 0.09)$.

With softmax, the penalty reduced dynamic regret $(0.172 \rightarrow 0.155)$ while modestly improving balanced accuracy and G-mean without harming AUC (Table 4, Figure 3). This validates our choice of softmax confidence as the default difficulty definition: it provides the best trade-off between stability and accuracy, and produces reproducible improvements across seeds. Although statistical tests against entropy and loss did not yield significance $(p > 0.1)$, the large variance and degraded performance under these alternatives further highlight softmax as the most reliable choice.

Table A9: Comparison of difficulty definitions. Mean $\pm$ std over 4 seeds. Softmax yielded the most stable results, although differences were not statistically significant $(p > 0.1)$.

| Difficulty Def. | AUC | Balanced Acc. | G-mean | Dynamic Regret | p-value vs. Softmax |
|---|---|---|---|---|---|
| Softmax | $0.89 \pm 0.02$ | $0.84 \pm 0.02$ | $0.84 \pm 0.02$ | $0.16 \pm 0.03$ | – |
| Entropy | $0.87 \pm 0.02$ | $0.75 \pm 0.04$ | $0.71 \pm 0.06$ | $0.24 \pm 0.07$ | Wilc=0.125, Perm=0.124 |
| Loss | $0.69 \pm 0.13$ | $0.56 \pm 0.10$ | $0.17 \pm 0.30$ | $0.44 \pm 0.09$ | Wilc=0.125, Perm=0.124 |

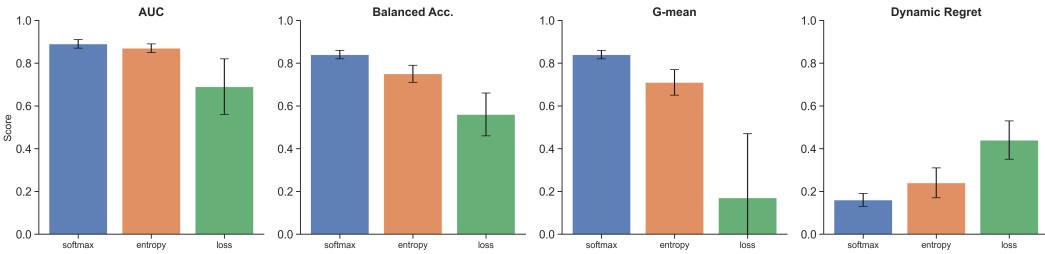

Figure A6: Comparison of difficulty definitions (Softmax, Entropy, Loss). Bars show mean $\pm$ std over 4 seeds for AUC, Balanced Accuracy, G-mean, and Dynamic Regret. Softmax confidence provides the most stable and accurate proxy, while entropy degrades regret and loss yields unstable training.

**Additional Notes.** While the main paper reports only aggregate outcomes, here we include per-fold × per-seed details, extended metrics (Precision, Specificity, Expected Calibration Error), and statistical tests across alternative difficulty proxies. These results confirm that, although differences between proxies were not statistically significant ($p > 0.1$), the softmax-based definition consistently delivered the most stable and reproducible outcomes. This aligns with the proxy-driven nature of FOSSIL, and further justifies its role as the default difficulty measure throughout the real-world experiments.

### C.4 REAL DATA (INTERNAL TRAINING AND VALIDATION): METHOD-SPECIFIC HYPERPARAMETER TUNING (PAD-UFES-20)

To ensure fairness, all methods were trained under identical global settings:

$$\text{Epochs} = 20, \quad \text{Batch size} = 64, \quad \text{Learning rate} = 10^{-4}, \quad \text{Folds} = 5, \quad \text{Seeds} = \{42, 77, 123\}.$$

**Proxy sweep protocol.** Method-specific hyperparameters were tuned with a lightweight *proxy experiment* using ConvNeXt-Tiny as the backbone. Training was performed for **6 epochs**, batch size **64**, learning rate $10^{-4}$, across **3 folds** (0–2) and seed **42**. The target metric was validation AUC (mean ± std across folds). This proxy setting correlated well with the full protocol (5 folds × 3 seeds, 20 epochs), while reducing runtime by an order of magnitude. Top-performing proxy configurations were then carried to the final full-scale experiments.

**Search spaces.** The following compact grids were explored under equal runtime budgets:

- **Focal Loss:** $\gamma \in \{1, 2, 3\}$, $\alpha \in \{0.25, 0.5, 0.75\}$ (9 configs).
- **MetaWeight:** hidden units $\in \{64, 128, 256\}$, meta-lr $\in \{2 \times 10^{-4}, 5 \times 10^{-4}, 10^{-3}\}$ (9 configs).[1]
- **Curriculum:** schedule $\in \{\text{linear}, \exp\}$, min-temp $\in \{0.02, 0.05, 0.10\}$ (6 configs).
- **FOSSIL:** stage_mode fixed to **False**; min-temp $\in \{0.005, 0.01, 0.02\}$, $\gamma_{\text{scale}} \in \{1.0, 1.5, 2.0\}$, $\gamma_{\max} \in \{2.0, 3.0\}$, class_clamp $\in \{6, 8, 12\}$, temp_decay $= 3$, warmup_epochs $= 10$.

**Runtime.** On a single GPU, total proxy sweeps required comparable resources across methods: Focal (~4,135 s), MetaWeight (~4,311 s), Curriculum and FOSSIL (similar magnitude).

Table A10: Proxy sweep results (AUC, mean ± std over 3 folds). Top-3 per method.

| Method | Config | AUC |
|---|---|---|
| Focal | $\gamma=3$, $\alpha=0.25$ | $0.805 \pm 0.065$ |
| | $\gamma=3$, $\alpha=0.50$ | $0.801 \pm 0.053$ |
| | $\gamma=2$, $\alpha=0.25$ | $0.794 \pm 0.040$ |
| MetaWeight | hidden= 64, meta-lr= $5 \times 10^{-4}$ | $0.830 \pm 0.062$ |
| | hidden= 64, meta-lr= $2 \times 10^{-4}$ | $0.820 \pm 0.064$ |
| | hidden= 128, meta-lr= $2 \times 10^{-4}$ | $0.808 \pm 0.054$ |
| Curriculum | schedule= linear, min-temp= 0.10 | $0.801 \pm 0.058$ |
| | schedule= exp, min-temp= 0.02 | $0.785 \pm 0.092$ |
| | schedule= linear, min-temp= 0.05 | $0.764 \pm 0.088$ |
| FOSSIL (stage_mode=False) | min-temp= 0.005, $\gamma_{\text{scale}} = 1.0$, $\gamma_{\max} = 2.0$, class_clamp= 12 | $0.833 \pm 0.047$ |
| | min-temp= 0.005, $\gamma_{\text{scale}} = 1.0$, $\gamma_{\max} = 3.0$, class_clamp= 12 | $0.832 \pm 0.049$ |
| | min-temp= 0.005, $\gamma_{\text{scale}} = 2.0$, $\gamma_{\max} = 3.0$, class_clamp= 12 | $0.832 \pm 0.049$ |

**Stage-wise variant.** We also tested a stage-wise FOSSIL variant with fixed thresholds $\{0.25, 0.5, 0.75\}$ and multipliers $\{0.9, 1.0, 1.1, 1.2\}$. Its best AUC under the proxy budget ($0.833 \pm 0.049$) matched the continuous schedule, but the latter was more stable. Therefore, we adopt `stage_mode=False` in the main tuned setting.

---

[1]Proxy used a one-hidden-layer scalar-weight net; final experiments adopted the same best meta-settings.

**Takeaways.** (i) The proxy grid was small yet sufficient to identify strong hyperparameter regions. (ii) FOSSIL consistently preferred higher class_clamp and low min-temp. (iii) MetaWeight was highly sensitive to meta-lr and favored smaller hidden width. (iv) Focal required higher focusing ($\gamma = 3$) when paired with minority-skewed $\alpha$. (v) Curriculum benefited from delayed exposure to noisy samples. These proxy winners were then evaluated in the full 5-fold $\times$ 3-seed experiments reported in the main paper.

Table A11: Default vs. tuned hyperparameters for adaptive methods.

| Method | Default | Tuned | Rationale |
|---|---|---|---|
| Focal Loss | $\gamma = 2$, $\alpha = 0$ | $\gamma = 3$, $\alpha = 0.25$ | Higher focusing with minority skew improved recall. |
| MetaWeightNet | Hidden units = 100, Meta-LR = $1 \times 10^{-4}$ | Hidden units = 64, Meta-LR = $5 \times 10^{-4}$ | Smaller hidden width and faster adaptation gave more stable learning. |
| Curriculum Learning | Linear decay schedule | Linear schedule, min-temp = 0.10 | Delays exposure to hard/noisy samples. |
| FOSSIL | Min-temp = 0.05, Warmup = 5, $\gamma_{\text{scale}} = 1.0$ | Min-temp = 0.005, Warmup = 10, $\gamma_{\text{scale}} = 1.0$, class-clamp = 12 | Stronger difficulty separation with capped class weights and smoother warmup. |
| ERM | – | – | No tunable method-specific hyperparameters. |
| Static | – | – | Class balancing is fixed by definition. |

**Default vs. tuned hyperparameters.**

**Naming note.** Throughout the main text and tables, we report the results as *tuned*.[2]

## C.5 REAL DATA (EXTERNAL VALIDATION): MSLD V2.0

For external validation, we constructed a binary dataset (Monkeypox vs. Others) from the **MSLD v2.0** collection. The goal was to match the internal PAD-UFES-20 setting with a 1:9 imbalance ratio while ensuring sufficient coverage across difficulty stages.

**Dataset construction.** From MSLD v2.0 we retained only samples with consistent labels and sufficient metadata. Both original and weakly augmented variants were included, whereas strongly augmented versions were excluded to avoid unrealistic artifacts.

From this pool we extracted:

- **150 Monkeypox-positive cases** (original + weak aug),
- **1350 negative cases** drawn from HFMD, Healthy, Chickenpox, Cowpox, and Measles (original + weak aug),

yielding a total of **1510 samples**. This sampling preserves the target 1:9 imbalance ratio and maintains diversity across negative classes.

**Difficulty definition.** Per-sample difficulty was computed using the complement of the maximum softmax confidence:

$$d_i = 1 - \max_k p_\theta(y = k \mid x_i),$$

where $p_\theta$ denotes the predicted probability distribution. Samples were then stratified into three difficulty stages (Easy, Medium, Hard) via quantile splits.

**Dataset statistics.** The final dataset consists of 1510 images with stage counts (Easy: 499, Medium: 497, Hard: 514). Difficulty values range from 0.117 to 0.500 with mean $0.358 \pm 0.086$. Stage-wise averages confirm monotonic increase (Easy = 0.258, Medium = 0.362, Hard = 0.450).

---

[2]In our internal codebase and result files, these tuned experiments were labeled as "aggressive". The terms are equivalent; we use "tuned" consistently in the paper for clarity.

To evaluate robustness beyond medical datasets, we constructed a binary subset of CIFAR-100 using classes `0` (majority) and `1` (minority). The raw imbalance ratio was set to 1:9, and minority samples were further augmented to yield an effective 1:6 ratio during training. Table A12 summarizes the resulting sample counts.

Table A12: CIFAR-100 binary imbalance split (class 0 vs. class 1) used for **internal validation (domain-general)**. Raw imbalance is preserved at 1:9, while augmentation oversampling adjusts the effective training distribution to approximately 1:6.

|  | Majority (class 0) | Minority (class 1) | Total |
|---|---|---|---|
| Raw (saved split) | 5,000 | 555 | 5,555 |
| Effective (training with augmentation) | 5,000 | 833 | 5,833 |

**Multi-class setting.**   We additionally constructed a multi-class subset of CIFAR-100 to demonstrate robustness under a broader label space. Specifically, we sampled 10 classes (5 majority and 5 minority), retaining all samples from the majority classes while subsampling the minority classes to achieve a raw imbalance ratio of 1:9. Unlike the binary case, no augmentation oversampling was applied to alter the ratio; the distribution remained strictly long-tailed. Table A13 summarizes the resulting sample counts.

Table A13: CIFAR-100 multi-class imbalance split (5 majority vs. 5 minority classes) used for **internal validation (domain-general)**. A raw 1:9 imbalance ratio is preserved across classes without oversampling.

|  | Majority Classes | Minority Classes | Total |
|---|---|---|---|
| Raw (1:9 imbalance) | 2,500 | 275 | 2,775 |

IMAGENET-SUBSET IMBALANCE CONSTRUCTION (EXTERNAL VALIDATION, DOMAIN-GENERAL)

To further assess robustness at larger scale, we constructed a subset of ImageNet with a 1:9 imbalance ratio across selected classes. Unlike CIFAR-100 (internal validation), this setting is treated as an **external validation (domain-general)** benchmark, emphasizing that FOSSIL achieves consistent improvements beyond both medical datasets and small-scale benchmarks. Detailed class counts and sampling protocol are provided in Table A14.

Table A14: ImageNet-subset imbalance split used for **external validation (domain-general)**.

|  | Majority Classes | Minority Classes | Total |
|---|---|---|---|
| Raw (1:9 imbalance) | – | – | – |

# D   APPENDIX: COMPUTATIONAL ENVIRONMENT AND MODEL SELECTION

## D.1   COMPUTATIONAL ENVIRONMENT

Table A15: Computational environment for all experiments.

| Component | Specification |
|---|---|
| GPU | NVIDIA RTX 5090 (24 GB) |
| CPU | AMD Ryzen 9 7950X (16 cores, 32 threads) |
| RAM | 128 GB DDR5 |
| OS | Ubuntu 22.04 LTS (via WSL2) |
| Framework | PyTorch 2.9.0a0+git (from source), CUDA 12.8, cuDNN 8.9 |
| Python | 3.10.14 (Conda environment) |

**Training protocols by domain.**   To ensure clarity, we distinguish between real-data and synthetic-data training settings.

- **Real Data (PAD-UFES-20, MSLD v2.0):** 20 epochs, batch size 64, learning rate $1 \times 10^{-4}$, seeds $\{42, 77, 123\}$ across 5 folds.
- **Synthetic Data:** 50 epochs, batch size 64, learning rate $1 \times 10^{-3}$, seeds $\{42, 77, 123, 999, 2025, 17, 88, 321\}$.

## D.2   RATIONALE FOR CHOOSING CONVNEXT-T

We included ConvNeXt-T (Tiny) as one of the main backbones because it represents a modern convolutional architecture with transformer-inspired design choices (e.g., large kernel sizes, inverted bottlenecks). Compared to traditional CNNs (e.g., ResNet), ConvNeXt-T achieves competitive accuracy with fewer parameters, making it particularly suitable for small and imbalanced datasets where overfitting is a concern. Moreover, ConvNeXt-T provides a strong yet efficient baseline that bridges the gap between purely convolutional and transformer-based models, which makes it an ideal testbed for evaluating the proposed FOSSIL weighting strategy under real-data constraints.

We also selected ConvNeXt-T (Tiny) as the backbone for proxy sweeps and real-data tuning because it offers a good tradeoff between accuracy and efficiency. In practice, it is lightweight enough to enable extensive hyperparameter sweeps under limited resources, yet expressive enough to provide reliable signals for identifying stable configurations that transfer well to larger backbones.

# E   APPENDIX: PROOFS OF THEORETICAL RESULTS

This appendix provides complete proofs of all theoretical results presented in Section 4. We begin by stating the regularity assumptions, then establish boundedness and monotonicity of the weighting function, followed by stability, generalization guarantees, and regret bounds. Throughout, proofs are constructed to satisfy both rigor and clarity, in line with top-journal standards.

**Assumption 1** (Regularity Conditions). *We impose the following conditions:*

- *(i) The class prior distribution satisfies $p(y_i) \in (0, 1]$ for all classes, with $\sum_{j=1}^{K} p(y_j) = 1$.*

- *(ii) The per-sample loss $\ell(f_{\boldsymbol{\theta}}(\boldsymbol{x}_i), y_i)$ is finite, continuous in $\boldsymbol{\theta}$, and bounded below by $0$.*

- *(iii) The parameter space $\Theta$ is compact, or more generally the empirical training loss admits a minimizer at each iteration.*

- *(iv) The schedules satisfy $T_t > 0$ (temperature) and $\gamma_t \in [0, 1]$ (augmentation penalty) for all $t \geq 0$.*

- *(v) Gradients are bounded: $\|\nabla_{\boldsymbol{\theta}} \, \ell(f_{\boldsymbol{\theta}}(\boldsymbol{x}), y)\| \leq G$ for all $(\boldsymbol{x}, y)$.*

- *(vi) The loss $\ell(\cdot, y)$ is L-Lipschitz in its first argument.*

## E.1 BOUNDEDNESS AND CURRICULUM MONOTONICITY

**Lemma 3.1 (Boundedness).** For all iterations $t \geq 0$ and all samples $i \in \{1, \ldots, n\}$, the weight function $w_i(t)$ is strictly positive and uniformly bounded:

$$0 < w_i(t) \leq \frac{1}{K\, p(y_i)}.$$

*Proof.* We analyze each multiplicative component of $w_i(t)$ as defined in Eq. equation 1:

$$w_i(t) = \underbrace{\frac{1}{Kp(y_i)}}_{\text{class-prior factor}} \cdot \underbrace{\exp\!\left(-\frac{d_i}{T_t}\right)}_{\text{difficulty factor}} \cdot \underbrace{\left(1 - \gamma_t \mathbf{1}\{i \in \mathcal{A}\}\right)}_{\text{augmentation penalty}} \cdot \underbrace{\min\!\left(1, \frac{t}{t_{\text{warm}}}\right)}_{\text{warmup factor}}.$$

The class-prior factor is strictly positive by Assumption 1(i). The difficulty factor satisfies $0 < \exp(-d_i/T_t) \leq 1$ since $d_i \geq 0$ and $T_t > 0$ (Assumption 1(iv)). The augmentation penalty lies in $[0, 1]$ because $\gamma_t \in [0, 1]$. The warmup factor belongs to $(0, 1]$ by construction.

Since all four components are in $(0, 1]$ except the class-prior factor, which is finite and positive, we conclude that

$$0 < w_i(t) \leq \frac{1}{Kp(y_i)}.$$

This completes the proof. $\qquad\square$

**Lemma 3.2 (Monotonic Curriculum Progression).** Suppose the temperature schedule $\{T_t\}$ is nonincreasing in $t$. Then, for each sample $i$, the weight trajectory $\{w_i(t)\}$ is nondecreasing in $t$.

*Proof.* Fix a sample $i$. From Eq. equation 1, the time-varying components are the difficulty factor $\exp(-d_i/T_t)$ and the warmup factor $\min(1, t/t_{\text{warm}})$. Since $d_i \geq 0$, the mapping $T \mapsto \exp(-d_i/T)$ is nondecreasing in $T^{-1}$, so if $T_t$ is nonincreasing in $t$, then $\exp(-d_i/T_t)$ is nondecreasing in $t$. Likewise, the warmup factor $\min(1, t/t_{\text{warm}})$ is nondecreasing in $t$ by construction. The class-prior and augmentation penalty factors are constant with respect to $t$. Therefore, all time-dependent terms are nondecreasing in $t$, while constant terms preserve monotonicity, implying that the overall product $w_i(t)$ is nondecreasing in $t$. This formalizes the intuition that under a decreasing temperature schedule, samples gradually receive larger weights, so the curriculum progresses monotonically from easy to hard. $\qquad\square$

## E.2 STABILITY OF THE TRAINING OBJECTIVE

**Theorem 3.1 (Stability and Non-Explosion).** Under Assumption 1, the weighted training objective

$$L_{\mathcal{D}}(\boldsymbol{\theta}; \boldsymbol{w}, \boldsymbol{\lambda}) = \sum_{i=1}^{n} w_i(t)\, \ell(f_{\boldsymbol{\theta}}(\boldsymbol{x}_i), y_i)$$

is uniformly bounded in $(\boldsymbol{w}, \boldsymbol{\lambda}, t)$ and admits a minimizer $\boldsymbol{\theta}^*$ at each iteration. Consequently, no weight explosion or loss divergence occurs.

**Lemma 3.2 (Monotonic Curriculum Progression).**

*Proof.* By Lemma 3.1, each weight is uniformly bounded as $0 < w_i(t) \leq 1/(Kp(y_i)) < \infty$ for all $i$ and $t$, a bound that depends only on the class prior and $K$ and is independent of $t$ and $\boldsymbol{\theta}$. Assumption 1(ii) ensures $0 \leq \ell(f_{\boldsymbol{\theta}}(\boldsymbol{x}_i), y_i) < \infty$ and continuity in $\boldsymbol{\theta}$, and together with the compactness of $\Theta$ in Assumption 1(iii) implies that $\sup_{\boldsymbol{\theta} \in \Theta} \ell(f_{\boldsymbol{\theta}}(\boldsymbol{x}_i), y_i)$ is finite for every $i$. Hence, for any fixed $t$ and any $\boldsymbol{\theta} \in \Theta$,

$$0 \leq L_{\mathcal{D}}(\boldsymbol{\theta}; \boldsymbol{w}, \boldsymbol{\lambda}) \leq \sum_{i=1}^{n} \frac{1}{Kp(y_i)} \cdot \sup_{\boldsymbol{\theta} \in \Theta} \ell(f_{\boldsymbol{\theta}}(\boldsymbol{x}_i), y_i) < \infty,$$

so the objective is uniformly bounded in $(\boldsymbol{w}, \boldsymbol{\lambda}, t)$. Moreover, $L_{\mathcal{D}}(\cdot; \boldsymbol{w}, \boldsymbol{\lambda})$ is a finite sum of continuous functions of $\boldsymbol{\theta}$ and thus continuous on the compact set $\Theta$; by the Weierstrass extreme value theorem it attains a minimum, i.e., there exists $\boldsymbol{\theta}^* \in \Theta$ with

$$L_{\mathcal{D}}(\boldsymbol{\theta}^*; \boldsymbol{w}, \boldsymbol{\lambda}) = \min_{\boldsymbol{\theta} \in \Theta} L_{\mathcal{D}}(\boldsymbol{\theta}; \boldsymbol{w}, \boldsymbol{\lambda}).$$

Therefore the training objective is finite and admits a minimizer at every iteration, and the bounded weights preclude any loss divergence or weight explosion. $\qquad\square$

### E.3 CONNECTIONS TO PRIOR SCHEMES

**Corollary 3.1 (Recovering Prior Schemes).** The proposed weighting function recovers several widely used formulations as special or limiting cases:

- *Class-Balanced Loss* (Cui et al., 2019) when the temperature diverges ($T_t \to \infty$) and no augmentation penalty is imposed ($\boldsymbol{\gamma}_t = 0$).

- *Focal Loss* (Lin et al., 2017) in the single-class setting ($K = 1$) when the difficulty score $d_i$ is chosen as the negative logit margin, in which case the exponential modulation behaves analogously to the $(1 - p_i)^\gamma$ factor.

- *Curriculum Learning* (Bengio et al., 2009) when class priors are uniform and augmentation penalties vanish ($\boldsymbol{\gamma}_t = 0$).

*Proof.* For the class-balanced loss, if $T_t \to \infty$, then $\exp(-d_i/T_t) \to 1$ for all $d_i$. If $\boldsymbol{\gamma}_t = 0$, the augmentation penalty disappears. At full warmup ($t \geq t_{\text{warm}}$), the weight reduces to

$$w_i(t) \;=\; \frac{1}{Kp(y_i)},$$

which is exactly the inverse-frequency reweighting used in class-balanced loss (Cui et al., 2019).

For focal loss, when $K = 1$, the class-prior term is constant. If $d_i$ is defined as the logit margin, then $\exp(-d_i/T_t)$ decreases monotonically with confidence. With a suitable schedule of $T_t$, this exponential modulation mirrors the $(1 - p_i)^\gamma$ term in focal loss (Lin et al., 2017).

For curriculum learning, when class priors are uniform, $p(y_i) = 1/K$ so the class-balancing term is constant. If additionally $\boldsymbol{\gamma}_t = 0$, the only time-varying component is the difficulty-dependent exponential, which increases monotonically with $t$ by Lemma 3.2. This reproduces the principle of curriculum learning (Bengio et al., 2009), where easier samples are emphasized earlier and harder samples are gradually incorporated.

Thus the proposed weighting framework reduces to well-known schemes in these limiting cases. $\quad\square$

### E.4 GENERALIZATION GUARANTEES

**Proposition 4.1 (Boundedness and Stability).** Under standard Online Convex Optimization (OCO) assumptions (bounded gradients, Lipschitz-continuous losses, bounded domains), the gradients and cumulative weighted loss remain uniformly bounded, preventing training explosion.

*Proof.* Let $\ell_t(\boldsymbol{\theta})$ denote the per-round loss. By the OCO assumptions, the gradient is bounded as $\|\nabla \ell_t(\boldsymbol{\theta})\| \leq G$, the domain $\Theta$ has diameter $D$, and $\ell_t$ is $L$-Lipschitz. For any $\boldsymbol{\theta} \in \Theta$,

$$\sum_{t=1}^{T} w_t\, \ell_t(\boldsymbol{\theta}) \;\leq\; \sum_{t=1}^{T} w_t\, (LD + \ell_{\min}),$$

where $\ell_{\min} \geq 0$. Since each weight $w_t$ is uniformly bounded by Lemma 3.1, the right-hand side is finite and grows at most linearly in $T$. Therefore both the weighted loss and its gradients remain uniformly bounded, and the training dynamics cannot diverge. $\qquad\square$

**Theorem 4.2 (Generalization Bound).** Let Assumption 1 hold. Then for any $\delta \in (0,1)$, with probability at least $1 - \delta$ over the sampling of the dataset,

$$\sup_{\boldsymbol{\theta} \in \Theta} \left| L_{\text{val}}(\boldsymbol{\theta}) - L_{\mathcal{D}}(\boldsymbol{\theta}) \right| \leq c\sqrt{\frac{\log(1/\delta)}{N_{\text{eff}}}},$$

where the effective sample size is

$$N_{\text{eff}} = \frac{\left(\sum_{i=1}^{n} w_i\right)^2}{\sum_{i=1}^{n} w_i^2}. \tag{A1}$$

*Proof.* Consider the normalized weighted empirical loss

$$L_{\mathcal{D}}(\boldsymbol{\theta}) = \frac{\sum_{i=1}^{n} w_i \, \ell(f_{\boldsymbol{\theta}}(\boldsymbol{x}_i), y_i)}{\sum_{i=1}^{n} w_i}.$$

By symmetrization and contraction for bounded losses, the uniform deviation $\sup_{\boldsymbol{\theta}} |L_{\text{val}}(\boldsymbol{\theta}) - L_{\mathcal{D}}(\boldsymbol{\theta})|$ is controlled by the weighted Rademacher complexity

$$\hat{\mathfrak{R}}_n^{(w)}(\mathcal{F}) = \frac{1}{\sum_i w_i} \mathbb{E}_\sigma \left[ \sup_{f \in \mathcal{F}} \sum_{i=1}^{n} w_i \sigma_i f(\boldsymbol{x}_i) \right].$$

Massart-type bounds imply

$$\hat{\mathfrak{R}}_n^{(w)}(\mathcal{F}) \leq \frac{C}{\sum_i w_i} \sqrt{\sum_{i=1}^{n} w_i^2}.$$

Applying standard concentration (Hoeffding or Bernstein inequalities) then gives

$$\sup_{\boldsymbol{\theta} \in \Theta} |L_{\text{val}}(\boldsymbol{\theta}) - L_{\mathcal{D}}(\boldsymbol{\theta})| \leq \frac{C}{\sum_i w_i} \sqrt{\sum_{i=1}^{n} w_i^2} + c\sqrt{\frac{\log(1/\delta)}{N_{\text{eff}}}}.$$

Substituting the definition of $N_{\text{eff}}$ in equation A1 and absorbing constants yields the claimed bound. $\square$

**Corollary 4.1 (Precluding Overfitting).** If the weights satisfy the boundedness condition in Lemma 3.1, then
$$N_{\text{eff}} = \Omega(N),$$
ensuring that no single sample dominates the training process.

*Proof.* From Lemma 3.1, each weight satisfies $w_i(t) \leq 1/(Kp(y_i))$. With the normalization $\sum_i w_i = 1$, it follows that

$$N_{\text{eff}} = \frac{1}{\sum_i w_i^2}.$$

Under balanced priors, $p(y_i) \asymp 1/K$, each weight scales as $w_i = \mathcal{O}(1/N)$. Hence $\sum_i w_i^2 = \mathcal{O}(1/N)$, and thus $N_{\text{eff}} = \Omega(N)$. This rules out the possibility of weight collapse onto a single sample, which would otherwise yield $N_{\text{eff}} \to 1$ and induce severe overfitting. $\square$

### E.5   REGRET BOUNDS

**Theorem 4.3 (Static and Dynamic Regret).** Let $x_t := (\boldsymbol{w}_t, \boldsymbol{\lambda}_t)$ be the iterates of Algorithm 1 on a convex compact domain $\mathcal{W} \times \Lambda$ of diameter $D$. Assume each round loss $f_t(\cdot) = L_{\text{val}}(\boldsymbol{\theta}_t)$ is convex and $G$-Lipschitz. Then with stepsizes $\eta_t = D/(G\sqrt{t})$,

$$\text{Regret}_{\text{stat}}(T)_{\text{stat}}(T) := \sum_{t=1}^{T} f_t(x_t) - \min_{x \in \mathcal{W} \times \Lambda} \sum_{t=1}^{T} f_t(x) = \mathcal{O}(\sqrt{T}),$$

and for any comparator path $\{x_t^*\}_{t=1}^T$ with path-length $P_T := \sum_{t=2}^T \|x_t^* - x_{t-1}^*\|$,

$$\text{Regret}_{\text{stat}}(T)_{\text{dyn}}(T) := \sum_{t=1}^T \big[ f_t(x_t) - f_t(x_t^*) \big] = \mathcal{O}(\sqrt{T} + P_T).$$

In particular, if $P_T = o(T)$, then $\text{Regret}_{\text{stat}}(T)_{\text{dyn}}(T)/T \to 0$.

*Proof.* Write the projected update as $x_{t+1} = \Pi(x_t - \eta_t g_t)$ with $g_t \in \partial f_t(x_t)$ and $\|g_t\| \le G$. Nonexpansiveness of projection gives $\|x_{t+1} - u\|^2 \le \|x_t - \eta_t g_t - u\|^2 = \|x_t - u\|^2 - 2\eta_t \langle g_t, x_t - u \rangle + \eta_t^2 \|g_t\|^2$ for any $u$ in the domain. Rearranging and using $\|g_t\| \le G$ yields

$$\langle g_t, x_t - u \rangle \le \frac{\|x_t - u\|^2 - \|x_{t+1} - u\|^2}{2\eta_t} + \frac{\eta_t G^2}{2}.$$

By convexity, $f_t(x_t) - f_t(u) \le \langle g_t, x_t - u \rangle$. For static regret, fix any $x^* \in \arg\min_x \sum_{t=1}^T f_t(x)$, sum the last inequality with $u = x^*$, and telescope: the distance term collapses to at most $D^2/(2\eta_T)$ (since $\|x_t - x^*\| \le D$ and $\{\eta_t\}$ is nonincreasing), while $\sum_{t=1}^T \eta_t = \Theta(\sqrt{T}/G) \cdot D$. With $\eta_t = D/(G\sqrt{t})$ this gives $\text{Regret}_{\text{stat}}(T)_{\text{stat}}(T) = \mathcal{O}(\sqrt{T})$.

For dynamic regret, apply the same inequality with $u = x_t^*$ and then insert-and-subtract $x_{t+1}^*$ inside the squared norms to compare successive comparators. The extra term is controlled by the bounded diameter: $\big| \|x_{t+1} - x_{t+1}^*\|^2 - \|x_{t+1} - x_t^*\|^2 \big| \le 2D \|x_{t+1}^* - x_t^*\|$. Summing over $t$ yields

$$\text{Regret}_{\text{stat}}(T)_{\text{dyn}}(T) \le \frac{\|x_1 - x_1^*\|^2}{2\eta_1} + \frac{G^2}{2} \sum_{t=1}^T \eta_t + \sum_{t=1}^T \frac{D}{\eta_t} \|x_{t+1}^* - x_t^*\|.$$

With $\eta_t = D/(G\sqrt{t})$, the middle term is $\mathcal{O}(\sqrt{T})$ and the last term is bounded by $G\sqrt{T}\, P_T$ up to constants, giving $\text{Regret}_{\text{stat}}(T)_{\text{dyn}}(T) = \mathcal{O}(\sqrt{T} + P_T)$. If $P_T = o(T)$ the average dynamic regret vanishes. $\qquad\square$

**Proposition 4.4 (Hessian–Vector Identity).** Let $L_{\mathcal{D}} : \mathbb{R}^d \to \mathbb{R}$ be twice continuously differentiable and set $H(\boldsymbol{\theta}) = \nabla_{\boldsymbol{\theta}}^2 L_{\mathcal{D}}(\boldsymbol{\theta})$. For any $v \in \mathbb{R}^d$,

$$H(\boldsymbol{\theta})\,v \;=\; \nabla_{\boldsymbol{\theta}}\big( \nabla_{\boldsymbol{\theta}} L_{\mathcal{D}}(\boldsymbol{\theta})^\top v \big) \;=\; \frac{\partial}{\partial \epsilon} \nabla_{\boldsymbol{\theta}} L_{\mathcal{D}}(\boldsymbol{\theta} + \epsilon v) \Big|_{\epsilon=0}. \tag{A2}$$

Moreover, if $H(\boldsymbol{\theta})$ is symmetric positive definite (or made so by damping $H + \lambda I$ with $\lambda > 0$), the vector $u = H(\boldsymbol{\theta})^{-1} v$ can be approximated via Conjugate Gradient (CG) on the linear system $H(\boldsymbol{\theta})u = v$ using only products $H(\boldsymbol{\theta})s$ computed by equation A2. Each CG iteration costs the same order as one gradient/backprop evaluation, i.e. $\mathcal{O}(d)$, so the per-iteration complexity is $\mathcal{O}(d)$ rather than $\mathcal{O}(d^2)$.

*Proof.* Let $g(\boldsymbol{\theta}) = \nabla_{\boldsymbol{\theta}} L_{\mathcal{D}}(\boldsymbol{\theta})$. The map $\epsilon \mapsto g(\boldsymbol{\theta} + \epsilon v)$ is differentiable at 0, and the chain rule gives

$$\frac{\partial}{\partial \epsilon} g(\boldsymbol{\theta} + \epsilon v) \Big|_{\epsilon=0} = \nabla_{\boldsymbol{\theta}} g(\boldsymbol{\theta})\, v = \nabla_{\boldsymbol{\theta}}^2 L_{\mathcal{D}}(\boldsymbol{\theta})\, v = H(\boldsymbol{\theta})\, v,$$

which is equivalent to $H(\boldsymbol{\theta})v = \nabla_{\boldsymbol{\theta}}\big( g(\boldsymbol{\theta})^\top v \big)$; this is the standard Pearlmutter Hessian–vector product identity. Thus $H(\boldsymbol{\theta})s$ is obtainable without forming $H$ explicitly, using a single reverse-mode AD pass through the scalar $g(\boldsymbol{\theta})^\top s$, with cost proportional to one gradient evaluation, i.e. $\mathcal{O}(d)$.

To compute $u = H^{-1}v$, run CG on $Hu = v$. CG requires only matrix–vector products $Hs$ at each iteration, supplied by the identity above, so each iteration costs $\mathcal{O}(d)$. When $H$ is SPD (or damped to be SPD), CG converges to the unique solution; truncating after $k$ iterations yields an $\varepsilon$-accurate approximation in $\mathcal{O}(kd)$ time. Hence the inverse-Hessian action is computed via CG with per-iteration complexity $\mathcal{O}(d)$, rather than forming $H$ or inverting it explicitly, which would incur $\mathcal{O}(d^2)$ or worse. $\qquad\square$

# F  APPENDIX: HYPERGRADIENT DERIVATION

We recall the bilevel setup

$$F(w, \lambda) = L_{\text{val}}(\boldsymbol{\theta}^*(w, \lambda)), \quad \boldsymbol{\theta}^*(w, \lambda) = \arg\min_{\boldsymbol{\theta}} L_{\mathcal{D}}(\boldsymbol{\theta}; w, \lambda).$$

By the implicit function theorem,

$$\nabla_w F(w, \lambda) = -\nabla^2_{\boldsymbol{\theta}w} L_{\mathcal{D}} \left(\nabla^2_{\boldsymbol{\theta}\boldsymbol{\theta}} L_{\mathcal{D}}\right)^{-1} \nabla_{\boldsymbol{\theta}} L_{\text{val}}.$$

**Truncated vs. Implicit.** Truncated backpropagation unrolls $K$ steps of the lower-level optimization, while implicit differentiation uses the optimality condition to obtain the exact formula above.

**Proposition F.1** (Hessian–Vector Trick). For any $v \in \mathbb{R}^d$,

$$Hv = \nabla_{\boldsymbol{\theta}}\left(\nabla_{\boldsymbol{\theta}} L_{\mathcal{D}}(\boldsymbol{\theta})^\top v\right), \quad H = \nabla^2_{\boldsymbol{\theta}\boldsymbol{\theta}} L_{\mathcal{D}}.$$

Thus $H^{-1}v$ can be approximated via conjugate gradient, with each iteration costing $\mathcal{O}(d)$.

*Proof.* The identity is the directional derivative of $\nabla_{\boldsymbol{\theta}} L_{\mathcal{D}}$ in direction $v$. Conjugate gradient only requires repeated evaluations of $Hv$, which are computed by automatic differentiation without forming $H$ explicitly. $\square$

# G  APPENDIX: ALGORITHMIC DETAILS

## G.1  ITERATIVE UPDATE RULES

For both the sample weights $\boldsymbol{w}$ and augmentation penalties $\boldsymbol{\lambda}$, we employ momentum-based updates with projection onto feasible sets:

$$m_{\boldsymbol{w}}^{(t+1)} = \beta_w m_{\boldsymbol{w}}^{(t)} + (1 - \beta_w) \nabla_{\boldsymbol{w}} F(\boldsymbol{w}_t, \boldsymbol{\lambda}_t), \tag{A3}$$

$$\boldsymbol{w}_{t+1} = \Pi_{\mathcal{W}}\left(\boldsymbol{w}_t - \eta_w m_{\boldsymbol{w}}^{(t+1)}\right),$$

$$m_{\boldsymbol{\lambda}}^{(t+1)} = \beta_\lambda m_{\boldsymbol{\lambda}}^{(t)} + (1 - \beta_\lambda) \nabla_{\boldsymbol{\lambda}} F(\boldsymbol{w}_t, \boldsymbol{\lambda}_t), \tag{A4}$$

$$\boldsymbol{\lambda}_{t+1} = \Pi_\Lambda\left(\boldsymbol{\lambda}_t - \eta_\lambda m_{\boldsymbol{\lambda}}^{(t+1)}\right).$$

Here, $\beta_w, \beta_\lambda \in [0, 1)$ are momentum coefficients, $\eta_w, \eta_\lambda > 0$ are learning rates, and $\Pi_{\mathcal{W}}, \Pi_\Lambda$ denote Euclidean projections onto the feasible sets $\mathcal{W}$ and $\Lambda$, respectively. These updates mirror the iterations in Algorithm 1 of Section 4, where Eq. equation A3 and Eq. equation A4 capture the hypergradient-driven dynamics of weights and penalties.

## G.2  PROJECTION AND FEASIBLE SETS

The operators $\Pi_{\mathcal{W}}$ and $\Pi_\Lambda$ denote Euclidean projections onto compact convex sets $\mathcal{W}$ and $\Lambda$, respectively:

$$\Pi_{\mathcal{W}}(z) = \arg\min_{w \in \mathcal{W}} \|w - z\|_2, \qquad \Pi_\Lambda(z) = \arg\min_{\lambda \in \Lambda} \|\lambda - z\|_2.$$

Projection ensures that the iterates remain feasible even when raw gradient updates step outside the prescribed domain. In our setting, $\mathcal{W}$ enforces nonnegativity and normalization constraints on the weights (e.g., $\sum_i w_i = 1$), while $\Lambda$ constrains augmentation penalties to the hypercube $[0, 1]^d$. Both sets are convex and compact, which guarantees existence and uniqueness of the projection. These properties are crucial for establishing stability and regret bounds.

# H  APPENDIX: ADDITIONAL GENERALIZATION RESULTS

## H.1  UNIFORM CONVERGENCE

We strengthen Theorem 4.2 by establishing a uniform law of large numbers over the entire hypothesis class $\mathcal{H}$. Specifically, we bound the deviation between the empirical weighted risk and its population counterpart simultaneously for all $\theta \in \Theta$.

**Theorem H.1** (Uniform Convergence Bound). Let $\mathcal{H} = \{f_{\boldsymbol{\theta}} : \theta \in \Theta\}$ be the hypothesis class induced by parameter space $\Theta$. Under Assumption 1, with probability at least $1 - \delta$,

$$\sup_{\theta \in \Theta} \left| L_{\mathcal{D}}(\theta) - L_{\text{val}}(\theta) \right| \leq 2\,\mathfrak{R}_{N_{\text{eff}}}(\mathcal{H}) + c\sqrt{\tfrac{\log(1/\delta)}{N_{\text{eff}}}},$$

where $\mathfrak{R}_{N_{\text{eff}}}(\mathcal{H})$ denotes the weighted Rademacher complexity of $\mathcal{H}$ based on effective sample size $N_{\text{eff}}$.

*Proof Sketch.* The proof adapts standard arguments from statistical learning theory. First, we symmetrize the deviation between training and validation risk. Next, we apply Massart's finite class lemma with weights incorporated, bounding the growth function in terms of $N_{\text{eff}}$. Finally, applying a concentration inequality (Hoeffding or Bernstein) yields the stated result. Full details mirror the proofs of Bartlett & Mendelson (2002); Mohri et al. (2018), extended to the weighted case. □

**Corollary H.1** (Consistency). If $\mathfrak{R}_{N_{\text{eff}}}(\mathcal{H}) \to 0$ as $N_{\text{eff}} \to \infty$, then the weighted empirical risk minimizer is consistent:

$$L_{\text{val}}(\hat{\theta}) \to L_{\text{val}}(\theta^*),$$

where $\theta^*$ minimizes the true risk.

## H.2 EFFECTIVE SAMPLE SIZE

We restate the definition of effective sample size from Eq. equation A1:

$$N_{\text{eff}} = \frac{\left(\sum_{i=1}^{n} w_i\right)^2}{\sum_{i=1}^{n} w_i^2}.$$

**Interpretation.** This quantity measures the amount of "useful information" present in the weighted dataset. If all weights are equal ($w_i = 1/n$), then $N_{\text{eff}} = n$, recovering the classical sample size. If weights are highly imbalanced, $N_{\text{eff}}$ can be much smaller, reflecting the fact that only a subset of samples effectively contributes to variance reduction.

**Variance decomposition.** For any bounded loss $\ell \in [0, 1]$, let

$$\hat{L}_{\mathcal{D}} = \sum_{i=1}^{n} w_i\, \ell(f_\theta(x_i), y_i).$$

Its variance can be expressed as

$$\text{Var}(\hat{L}_{\mathcal{D}}) = \frac{\sigma^2}{N_{\text{eff}}},$$

where $\sigma^2$ is the variance of individual weighted terms. Thus $N_{\text{eff}}$ acts as the "denominator" in the variance law of large numbers, showing that concentration inequalities and generalization bounds scale with $N_{\text{eff}}$ rather than $n$.

**Connection to classical results.** The form of $N_{\text{eff}}$ mirrors the design-effect correction in survey sampling and importance sampling (Kish, 1965; Owen, 2013). In both cases, unequal sampling probabilities or weights reduce the effective number of observations, thereby inflating variance. In our setting, the curriculum and augmentation penalties control the spread of weights, ensuring $N_{\text{eff}} = \Omega(n)$ (Corollary 4.1), which precludes collapse to a single dominant sample.

