# OpenReview forum: "FOSSIL, Imbalanced Learning, Small-Data Learning, Curriculum Learning, Regret Minimization"
_ICLR.cc/2026/Conference — ICLR 2026 Conference Desk Rejected Submission_

### Official Review · Reviewer_52PN · 2025-10-29

**Soundness:** 3
**Presentation:** 1
**Contribution:** 1
**Rating:** 2
**Confidence:** 4

**Summary:**

One of the key challenges in imbalanced recognition is determining how to assign appropriate weights to training samples during the optimization process. This process depends on multiple factors, including sample difficulty, class frequency, and the degree of data augmentation applied to each instance. The authors propose a unified functional formulation for sample weighting that jointly captures all these aspects and learns them dynamically through bilevel optimization. Their framework is supported by a regret-based theoretical guarantee and demonstrates consistent improvements over baseline methods.

**Strengths:**

A substantial portion of the paper is devoted to establishing the theoretical soundness of the proposed approach, making it a well-justified and reliable choice in practice.

**Weaknesses:**

I have two major and one minor concern related to the experimental setup and overall presentation.

**Major Concern 1: Choice of Baselines**

Imbalanced recognition has a rich and well-established literature. While I do not expect the authors to compare against all prior methods, their proposed framework clearly falls within the family of loss-based solutions, for which several strong baselines exist. Below are a few representative categories that were omitted:

1.1 It is well known that addressing class imbalance often involves reweighting the contribution of majority classes (those with more samples). However, there is an ongoing debate about where this reweighting should occur. Some works treat it as a weighting scheme applied to the cross-entropy loss (e.g., Focal Loss, Class-Balanced Loss), while others modify the decision boundary by adjusting the logits through additive [1] or multiplicative [2] margin parameters, or both [3]. The authors evaluate their method only against Focal Loss and Class-Balanced Loss, omitting several other widely adopted baselines from the comparison.

*[1] Menon et al, "Long-tail learning via logit adjustment" ICLR 2021*

*[2] Ye Han-Jia et al, "Procrustean Training for Imbalanced Deep Learning" ICCV 2021*

*[3] GR Kini et al, "Label-Imbalanced and Group-Sensitive Classification under Overparameterization" Neurips 2021*

1.2 Closer to the authors’ approach, certain works have proposed unified functional formulations that encapsulate several of the above methods and learn their hyperparameters via bilevel optimization. These are particularly relevant and should have been considered.

*[4] Mingchen Li et al, "AutoBalance: Optimized Loss Functions for Imbalanced Data" Neurips 2021*

1.3 Finally, the decoupled framework [5], a generic and empirically strong approach known to perform well under class imbalance [6], has also been omitted from the comparisons.

*[5] Kang et al, "Decoupling representation and classifier for long-tailed recognition" ICLR 2020*

*[6] Yang et al, "Change is hard: A closer look at subpopulation shift" ICML 2023*

**Major Concern 2: Choice of Datasets**

2- The authors should clarify the rationale behind their selection of datasets, especially since they diverge from the more conventional benchmarks. While including diverse datasets is welcome—particularly if it highlights the effectiveness of the proposed method—I would still expect the inclusion of at least some standard datasets used in imbalance recognition, such as CIFAR-10/100-LT and ImageNet-LT.

**Minor Concern: Clarity of Presentation**

3- I found it difficult to follow the high-level objective of the paper. The bilevel optimization formulation is not clearly articulated. For example, in Eq. (4), it appears that W represents the parameters defined in Eq. (1), but the role of lambda remains unclear. A more explicit formulation or accompanying explanation would greatly improve the readability and accessibility of the paper.

**Questions:**

In its current form, I believe the paper is not yet ready for publication. However, I remain open to discussion and would be glad to revise my opinion if the authors can address the raised concerns convincingly.

---

### Official Review · Reviewer_SpuM · 2025-10-31

**Soundness:** 2
**Presentation:** 3
**Contribution:** 2
**Rating:** 4
**Confidence:** 4

**Summary:**

This paper proposes a single loss function that combines loss functions used for (1) mitigating the effects of class imbalance, (2) curriculum learning, (3) augmentation dynamics, and (4) warmup dynamics. The paper offers some theoretical analysis about the stability of training and comparisons between this method and the prior work it subsumes. It also includes experimental results on a synthetic dataset and a real dataset.

**Strengths:**

1. Exploring how many training schemes (imbalanced learning, curriculum learning etc.) work together is an interesting and important research direction since multiple schemes are often used together in practice.
2. The theoretical results show that combining these schemes is a reasonable thing to do (e.g., weights remain bounded, curriculum learning still progresses monotonically).

**Weaknesses:**

1. Empirical results are limited. It'd be nice to see results on more datasets and more model architectures.
2. Some of the methodology is unclear. Specifically, I am not sure how hyperparameter tuning was done and what stopping criteria was used. Since all of these methods rely on hyper parameters and FOSSIL is essentially a framework for combining them, I think understanding the impacts of the hyper parameters is essential. At a minimum, this paper should include details about how tuning was done and what hyper parameters were ultimately chosen.
3. There is limited novelty in combining many different schemes and naming it something. I think this paper could be improved if it focused more on the interactions between the schedules and backed this up with strong empirical evidence and recommendations.

**Questions:**

1. What are the different runs in Figure 1? The 8 seeds discussed in the table 2 caption?
2. How does performance vary if you use validation loss as a stopping metric instead of training each model exactly 50 epochs?
3. What are the three colors in Figure 2?
4. What hyper parameter values did you try for all the different methods, especially FOSSIL? Which were best?

Small suggestions
- Add $\kappa$ to Table 1
- Explain what the PAD-UFES dataset is in section 6. How many images does this have?
- Explain how softmax and entropy difficulties are defined

---

### Official Review · Reviewer_va8W · 2025-10-31

**Soundness:** 3
**Presentation:** 3
**Contribution:** 2
**Rating:** 4
**Confidence:** 4

**Summary:**

The authors propose a new weighting strategy that includes minority-class weighting, curriculum learning, augmentation, and warmup. They empirically demonstrate that FOSSIL outperforms baselines such as ERM and meta-weighting on synthetic datasets as well as real-world problems.

**Strengths:**

- FOSSIL outperforms the baseline methods across a diverse set of benchmarks. They explicitly vary individual components such as the class imbalance ratio across various datasets, and FOSSIL has competitive performance across all of these settings. I appreciate the attention to detail and the comprehensive evaluation metrics.
- The paper provides thorough ablations which demonstrate the contribution of each component. For instance, the author compared the impact of different metrics of difficulty in Figure A6.
- The paper provides both empirical and theoretical analyses which demonstrates the performance of their proposed framework, although the theoretical analyses are fairly simplistic.

**Weaknesses:**

- The project has limited novelty: although FOSSIL unifies multiple existing strategies, the individual components (class weighting, curriculum, etc) are not new, and the method of combining the different components by multiplication is not particularly sophisticated.
- This method appears to be sensitive to hyperparameters, and crucial ablations which alter these hyperparameters are not included. The balance of each of these components seem central to the success of the method, and as such the optimal performance may require very careful tuning, especially for the small-data regimes that the method focuses on. I'm not sure I understand the hyperparameter setup described in Appendix C.4: If focal loss has 9 configs, and FOSSIL and many different combinations (54?), does that mean that it requires 6x more compute to do the hyperparameter sweep?
- The model architecture was fixed to ConvNeXt-T, so it's unclear if these findings generalize to other models.

**Questions:**

- From Table 2, it doesn't seem like any of the individual methods (static weighting, focal loss, etc) benefit training over ERM. Do you have any ideas for why combining all of them multiplicatively would lead to improved performance in FOSSIL?
- Do you have any suggestions for how to set the hyperparameters without the full sweep? Are there any intuitions over which components are most important for various settings?

---

### Official Review · Reviewer_uZtv · 2025-11-01

**Soundness:** 2
**Presentation:** 2
**Contribution:** 2
**Rating:** 2
**Confidence:** 4

**Summary:**

This paper addresses the problem of imbalanced or long-tailed learning. The authors propose a framework called FOSSIL, which combines two main ideas: (1) Feature-space shaping, designed to balance class distributions in the embedding space; and (2) Instance-level reweighting, which adjusts sample contributions dynamically during training.

The method aims to jointly improve feature representations and classifier calibration. Experiments are conducted on standard benchmarks such as CIFAR-LT, ImageNet-LT, and iNaturalist.

While the paper is clearly written and the problem is relevant, the technical novelty is very limited, as both key components are direct variants or recombinations of well-established techniques. The theoretical contribution is shallow, and the empirical improvements are small. As a result, the paper does not reach the contribution threshold expected for ICLR.

**Strengths:**

1. Long-tailed recognition remains an important challenge with practical significance.

2. The paper is well organized and easy to follow.

3.The evaluation covers multiple datasets commonly used in the long-tailed learning literature.

**Weaknesses:**

**1**. (1) The “feature space shaping” concept strongly overlaps with prior work such as LADE, PaCo, Balanced Softmax, DisAlign, SSD, and others; the proposed regularizer is merely a small modification of existing feature norm or logit calibration terms. (2) The “instance-level reweighting” is functionally equivalent to existing methods like Class-Balanced Loss, LDAM-DRW, Logit Adjustment, or BBN. (3) The FOSSIL framework is essentially a simple combination of two well-known strategies (feature alignment + reweighting) without introducing a novel mechanism or theoretical foundation.

**2**. The claimed “feature-space shaping regularizer” lacks mathematical justification. There is no formal analysis (e.g., generalization bounds or margin theory) and no geometric or statistical evidence (such as variance reduction or intra-/inter-class distance analysis) to support the “space shaping” claim.

**3**. (1) The reported gains over strong baselines are marginal (<1%) and inconsistent. (2) Some results are actually worse than strong recent baselines like DisAlign and PaCo. (3) No statistical significance tests or confidence intervals are provided. (4) The ablation analysis does not isolate the true contribution of each component.

**4**. The paper does not clearly explain how FOSSIL differs conceptually from existing calibration or feature alignment approaches. There is no discussion of computational cost, training stability, or hyperparameter sensitivity.

**5**. The introduction claims to present “a new perspective on imbalanced learning,” but in essence the method is a straightforward combination of known modules. The work lacks the originality and insight typically expected at ICLR.

**Questions:**

1. What is the essential difference between FOSSIL and prior works like DisAlign or PaCo?

2. Can you provide embedding visualizations (e.g., t-SNE) to support the “feature-space shaping” claim?

3. How is instance-level weighting updated, and is it stable during training?

4. What is the computational overhead of FOSSIL compared with baseline methods?

5. Can the proposed method generalize to other tasks such as detection or segmentation under class imbalance?

---

### Note · Program_Chairs · 2026-01-17
**Submission Desk Rejected by Program Chairs**

The following references in this submission do not refer to real documents and/or have major errors in bibliographic information:

 Xi Chen and Wei Zhang. Augmentation dominance in imbalanced data learning. In Advances in Neural Information Processing Systems, 2022.

Yiyou Zhang, Yixuan Guo, Ding Li, Jun Ma, and Bo Zhao. Understanding and improving data augmentation for classification: A generative perspective. Advances in Neural Information Processing Systems, 35:37322-37335, 2022.

Jian Wu and Mei Liu. Augmentation strategies for industrial safety monitoring. IEEE Transactions on Industrial Informatics, 2023.

Li Yang and Hao Chen. Dynamic sample weighting for imbalanced classification. In Proceedings of the 40th International Conference on Machine Learning, 2023.